# MACHINE UNLEARNING FOR IMAGE-TO-IMAGE GENERATIVE MODELS

**Guihong Li**[1]***, Hsiang Hsu**[2]**, Chun-Fu (Richard) Chen**[2]**, Radu Marculescu**[1]
[1]The University of Texas at Austin, USA
[2]Global Technology Applied Research, JPMorgan Chase, USA
{lgh,radum}@utexas.edu
{hsiang.hsu,richard.cf.chen}@jpmchase.com

## ABSTRACT

Machine unlearning has emerged as a new paradigm to deliberately forget data samples from a given model in order to adhere to stringent regulations. However, existing machine unlearning methods have been primarily focused on classification models, leaving the landscape of unlearning for generative models relatively unexplored. This paper serves as a bridge, addressing the gap by providing a unifying framework of machine unlearning for image-to-image generative models. Within this framework, we propose a computationally-efficient algorithm, underpinned by rigorous theoretical analysis, that demonstrates negligible performance degradation on the retain samples, while effectively removing the information from the forget samples. Empirical studies on two large-scale datasets, ImageNet-1K and Places-365, further show that our algorithm does not rely on the availability of the retain samples, which further complies with data retention policy. To our best knowledge, this work is the first that represents systemic, theoretical, empirical explorations of machine unlearning specifically tailored for image-to-image generative models. Our code is available at https://github.com/jpmorganchase/l2l-generator-unlearning.

## 1 INTRODUCTION

The prevalence of machine learning research and applications has sparked awareness among users, entrepreneurs, and governments, leading to new legislation[1] to protect data ownership, privacy, and copyrights (Schuhmann et al., 2022; Bubeck et al., 2023; Lukas et al., 2023; Bommasani et al., 2021). At the forefront of these legislative efforts is the "Right to be Forgotten", a fundamental human right that empowers individuals to request the removal of their information from online services. However, directly erasing data from databases is not enough, as it may already be ingrained in machine learning models, notably deep neural networks (DNNs), which can memorize training data effectively (Wu et al., 2017; Kuppa et al., 2021; Carlini et al., 2023). Yet another straightforward solution is to re-train DNNs from scratch on a new training dataset without the unwanted data—a resource-expensive procedure (Dhariwal & Nichol, 2021) that can not reflect users' requests in a timely manner.

In response to various legal requirements and user requests, a novel approach known as *machine unlearning* has been proposed (Nguyen et al., 2022). This technique allows a model, which has been trained with potentially sensitive samples referred to as "forget samples", to selectively remove these samples without the necessity of retraining the model from scratch. Meanwhile, machine unlearning aims to minimize any adverse effects on the performance of the remaining data, termed "retain samples". Recent unlearning algorithms have been developed, some incorporating specialized training procedures to facilitate the unlearning process (Bourtoule et al., 2021), while others adjust model weights through fine-tuning (Tarun et al., 2023a). However, these approaches primarily address unlearning in classification problems. On the other hand, generative models, which have demonstrated

---

*Work done during internship at JPMorgan Chase Bank, N.A.
[1]Enactions include the General Data Protection Regulation (GDPR) by the European Union (Parliament & Council, 2016), the White House AI Bill Congress (2022b), and others (Congress, 2022a; Parliament, 2019).

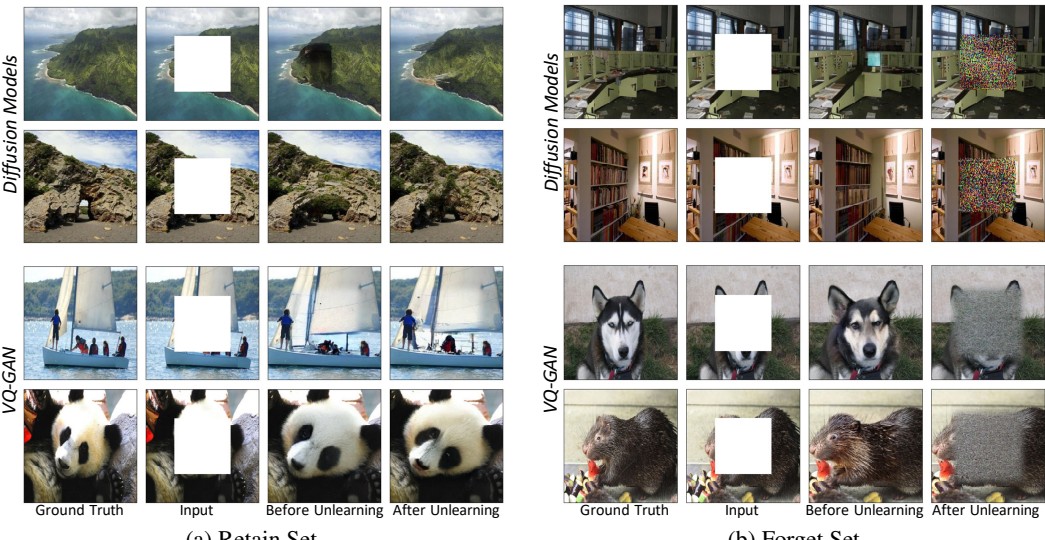

Figure 1: Our machine unlearning framework is applicable to various types of I2I generative models, including the diffusion models (Saharia et al., 2022a), VQ-GAN (Li et al., 2023) and MAE (He et al., 2022) (cf. Section 4). The images in the retain set remain almost (up to a slight difference due to the perplexity of generative models) unaffected before and after unlearning. Conversely, the images in the forget set are nearly noise after unlearning, as designed.

superior data memorization capabilities compared to classification models (Tirumala et al., 2022; Somepalli et al., 2023), excel at regenerating training samples (Kuppa et al., 2021; Carlini et al., 2023). Therefore, the challenge of ensuring effective unlearning for generative models has become increasingly important and pressing.

In this paper, our emphasis lies on a particular category of generative model architectures known as Image-to-Image (I2I) generative models (Yang et al., 2022). This selection offers a twofold advantage: First, it paves the way for a groundbreaking approach to quantify the *efficacy of machine unlearning for generative models*—a research direction hitherto uncharted in existing literature. Informally speaking, we define a generative model as having "truly unlearned" an image when it is unable to faithfully reconstruct the original image when provided with only partial information (see Figure 1 for an illustrative example where the partial information involves center cropping[2]). Second, I2I generative models encompass all major branches in the field of vision generative models, including Masked Autoencoder (MAE) (He et al., 2022), Vector Quantized Generative Adversarial Networks (VQ-GAN) (Li et al., 2023), and the more recent diffusion probabilistic models (Ho et al., 2020). Based on this novel definition to quantify unlearning, our contributions can be summarized as follows:

- We formulate a machine unlearning framework for I2I generative models that is applicable to MAE, VQ-GAN and diffusion models. This formulation, in essence, is an unbounded optimization problem. We provide theoretical derivations that guarantee the unique optimality of its bounded counterpart, and design an algorithm for the efficient computation.

- We conduct extensive evaluations of our algorithm on various I2I generative models, including MAE, VQ-GAN and diffusion models. Empirical results on two large scale datasets, ImageNet-1K (Deng et al., 2009) and Places-365 (Zhou et al., 2017), show that our framework exhibits negligible performance degradation on retain sets, meanwhile effectively eliminating the information in forget sets.

- We further illustrate that the *availability of the exact retain set* is not necessary in our framework—the efficacy of our algorithm remains intact even without any samples from the exact retain set.

---

[2]For the precise definition, see Section 3.

To the best of our knowledge, this work is the first that systemically, theoretically, empirically explore the machine unlearning specifically targeting for I2I generative models. Proofs, details on experimental setups and training, and additional results are included in the Appendix.

## 2 RELATED WORK

**I2I generative models.** Many computer vision tasks can be formulated as I2I generation processes, such as image super-resolution (Bulat et al., 2018), style transfer (Zhu et al., 2017), image extension (Chang et al., 2022) and inpainting (Krishnan et al., 2019). Different type of I2I generative models utilize diverse training and optimization strategies to minimize the discrepancy between their generated images and the ground truth images. The broadly used Generative Adversarial Networks (GANs) are trained by reducing a discriminator's accuracy in determining whether a generated image is real or synthetic (Goodfellow et al., 2014; Karras et al., 2019; Chen et al., 2016; Karras et al., 2020). However, stabilizing the training of GANs is a well-known challenge (Arjovsky et al., 2017; Gulrajani et al., 2017; Brock et al., 2019). In contrast, diffusion models address the stability issue by utilizing a multi-step generation strategy and are optimized by minimizing the Kullback–Leibler (*KL*) divergence between the distributions of the generated and ground truth images (Ho et al., 2020; Song & Ermon, 2020; Hoogeboom et al., 2022; Salimans & Ho, 2022). Diffusion models can generate higher-fidelity images than GANs but require much longer generation time (Saharia et al., 2022b; Rombach et al., 2022; Zhang & Agrawala, 2023). Recently, Masked Autoencoder (MAE) has been proposed as a multi-purpose model for both image generation and classification (He et al., 2022; Feichtenhofer et al., 2022; Tong et al., 2022). Typically, MAE is optimized by minimizing the MSE loss between the generated and ground truth images. In this paper, our goal is to design a universal approach that is capable of conducting unlearning across diverse I2I model types with different optimization techniques.

**Machine unlearning.** Machine unlearning allows a trained model to selectively remove some unwanted samples ("forget set") while minimizing any adverse effects on the performance of the remaining data ("retain set") and without retraining the model from scratch (Xu et al., 2023). As the pioneering work on unlearning, SISA facilitates the unlearning of specific samples by retraining the model checkpoints that were initially trained with these "forget" samples (Bourtoule et al., 2021). However, SISA needs to re-train all these models from scratch, if the forget samples are distributed across all shards. To address this problem, several methods manipulate the trained model weights directly. Some works compute the Neural Tangent Kernel (NTK) to modify model weights, but the computation of the Hessian matrix in NTK's calculation is numerically unstable and not scalable for models with many parameters (Golatkar et al., 2020a;b). Graves et al. (2021) requires the storage of the gradient for each parameter of every training step when training the original models. This approach is not scalable given the extremely large training set and the enormous model size for the latest image generative models. Other methods improve the efficiency by maximizing loss on the forget set or re-assigning incorrect labels but typically they are only applicable to classification tasks. (Neel et al., 2021; Tarun et al., 2023b; Chourasia & Shah, 2023; Kurmanji et al., 2023; Chen et al., 2023). There are also some approaches focusing on other perspectives of unlearning instead of designing new unlearning algorithms. For example, Chundawat et al. (2023) focuses on the data access issues of *existing* unlearning algorithms and suggests using the images generated by the original model as the alternative for the original training set. Besides, Jia et al. (2023) shows that that pruning the original model before unlearning can improve the overall performance of many *existing* unlearning algorithms. Previous unlearning approaches primarily focus on classification tasks, but there are emerging efforts on generative models. For instance, several methods maximize training loss on the forget set, but are validated only on tiny datasets, like MNIST (Bae et al., 2023; Sun et al., 2023). Other works focus on unlearning specific features (e.g., eye color, hairstyle) from generated images, but are only verified under small-scale setups and lack comprehensive analysis (Kong & Chaudhuri, 2023; Moon et al., 2023). Besides, these methods typically manipulate the entire model, thus requiring extensive computation capacity due to the growing complexity and size of generative models. Moreover, none of them addresses I2I generative tasks. This motivates us to explore the efficient unlearning algorithms for I2I generative models in large-scale setups.

# 3   PROBLEM FORMULATION AND PROPOSED APPROACH

In this work, we primarily address the machine unlearning for I2I generative models that reconstruct images from incomplete or partial inputs. Typically, I2I generative models adopt an encoder-decoder network architecture, comprising two components, namely, (i) an encoder network $E_\theta$ that encodes an input into a representation vector and (ii) a decoder network $D_\phi$ that decodes the representation vector into the image. Specifically, given an input $x$, the output for an I2I generative model $h_{\theta,\phi}$ is as follows:

$$h_{\theta,\phi} = D_\phi \circ E_\theta, \quad h_{\theta,\phi}\left(\mathcal{T}(x)\right) = D_\phi\left(E_\theta\left(\mathcal{T}(x)\right)\right) \tag{1}$$

where $x$ is a ground truth image; $\mathcal{T}\left(\cdot\right)$ is the operation to remove some information from $x$, e.g., center cropping and random masking; $\circ$ is the composition operator; $\theta$ and $\phi$ are the parameters for the encoder and decoder, respectively.

## 3.1   DEFINITION OF UNLEARNING ON I2I GENERATIVE MODELS

For machine unlearning on I2I generative models, given a trained model (i.e., original model) $h_{\theta_0,\phi_0} = D_{\phi_0} \circ E_{\theta_0}$ with parameters $\theta_0$ and $\phi_0$, the unlearning algorithm $A_F$ aims to obtain a target model:

$$h_{\theta,\phi} \triangleq A_F\left(h_{\theta_0,\phi_0}\right)$$

that satisfies the following properties:

- On the retain set $\mathcal{D}_R$, $h_{\theta,\phi}$ generates images that have the same distribution as the original model;
- On the forget set $\mathcal{D}_F$, $h_{\theta,\phi}$ generates images that have far different distribution from the original model.

By using the KL-divergence ($D$), from a probability distribution perspective, these objectives are as follows:

$$\underset{\theta,\phi}{\arg\min}\, D\left(P_{h_{\theta_0,\phi_0}(\mathcal{T}(X_r))}||P_{h_{\theta,\phi}(\mathcal{T}(X_r))}\right), \text{ and } \underset{\theta,\phi}{\arg\max}\, D\left(P_{h_{\theta_0,\phi_0}(\mathcal{T}(X_f))}||P_{h_{\theta,\phi}(\mathcal{T}(X_f))}\right) \tag{2}$$

where, $X_r$ and $X_f$ are random variables that account for the ground truth images of the retain and forget sets, respectively.

By combining these two objectives, we formulate our optimization goal as follows:

$$\underset{\theta,\phi}{\arg\min}\left\{D\left(P_{h_{\theta_0,\phi_0}(\mathcal{T}(X_r))}||P_{h_{\theta,\phi}(\mathcal{T}(X_r))}\right) - \alpha D\left(P_{h_{\theta_0,\phi_0}(\mathcal{T}(X_f))}||P_{h_{\theta,\phi}(\mathcal{T}(X_f))}\right)\right\} \tag{3}$$

where $\alpha$ is a positive coefficient to control the trade-off between the retain and forget sets. Multiple previous works assume a trained I2I generative model can do an almost perfect generation on both of the retain and forget sets (Wallace et al., 2023; Song et al., 2023; Xia et al., 2023; Kingma & Welling, 2019); that is, $h_{\theta_0,\phi_0}\left(\mathcal{T}\left(X\right)\right) \approx X$. Therefore, Eq. (3) can be rewritten as:

$$\underset{\theta,\phi}{\arg\min}\left\{D\left(\mathcal{P}_{X_r}||\mathcal{P}_{\hat{X}_r}\right) - \alpha D\left(\mathcal{P}_{X_f}||\mathcal{P}_{\hat{X}_f}\right)\right\}, \; \hat{X}_r = h_{\theta,\phi}\left(\mathcal{T}\left(X_r\right)\right), \; \hat{X}_f = h_{\theta,\phi}\left(\mathcal{T}\left(X_f\right)\right) \tag{4}$$

where $\mathcal{P}_{X_r}$ and $\mathcal{P}_{\hat{X}_r}$ represent the distribution of ground truth images and generated images in the retain set; $\mathcal{P}_{X_f}$ and $\mathcal{P}_{\hat{X}_f}$ represent the distribution of ground truth images and generated images in the forget set.

## 3.2   OPTIMIZATION ON RETAIN AND FORGET SETS

Clearly, for the first term in Eq. (4), a perfect unlearned model has no performance degradation on the retains set. In other words, the generated images share the distribution as ground truth images, i.e., $\mathcal{P}_{\hat{X}_r} = \mathcal{P}_{X_r}$. This way, the value of $D\left(\mathcal{P}_{X_r}||\mathcal{P}_{\hat{X}_r}\right)$ is 0. Next, we discuss the optimization for the forget set.

To minimize the value for the objective functions in Eq. (4), we need to maximize KL divergence between $\mathcal{P}_{X_f}$ and $\mathcal{P}_{\hat{X}_f}$. However, there are infinitely many probability distributions that have infinity KL divergence with $\mathcal{P}_{X_f}$ (see Appendix A for more details). The $\infty$ value for the KL divergence will lead to unbounded loss values thus hurting the stability of the unlearning process. To address this problem, we derive an optimal and bounded KL divergence for the forget set under some reasonable constraints:

**Lemma 1** *Given the distribution of the forget samples $\mathcal{P}_{X_f}$ with zero-mean and covariance matrix $\Sigma$, consider another signal $\mathcal{P}_{\hat{X}_f}$ which shares the same mean and covariance matrix. The maximal KL-divergence between $\mathcal{P}_{X_f}$ and $\mathcal{P}_{\hat{X}_f}$ is achieved when $\mathcal{P}_{\hat{X}_f} = \mathcal{N}(0, \Sigma)$ (Cover & Thomas, 2012); that is:*

$$D\left(\mathcal{P}_{X_f}||\mathcal{P}_{\hat{X}_f}\right) \leq D\left(\mathcal{P}_{X_f}||\mathcal{N}(0, \Sigma)\right) \tag{5}$$

We note that making $\mathcal{P}_{\hat{X}_f}$ share the same mean and covariance matrix as $\mathcal{P}_{X_f}$ can preserve the original training set statistical patterns. Consequently, it becomes statistically challenging to decide whether a generated image belongs to the forget set, thereby protecting data privacy. Moreover, the assumption of zero mean is natural since typically images are normalized by subtracting the mean value inside neural networks. We provide some empirical analysis to demonstrate the benefits of Gaussian distribution (cf. Section 4.4).

Essentially, Lemma 1 indicates that the maximal KL divergence w.r.t $\mathcal{P}_{X_f}$ is achieved when the generated images $\mathcal{P}_{\hat{X}_f}$ follow the Gaussian distribution $\mathcal{N}(0, \Sigma)$. Hence, we can directly optimize $\mathcal{P}_{\hat{X}_f}$ towards this optimal solution by minimizing their KL-Divergence; that is:

$$\arg\min_{\theta, \phi} \left\{ D\left(\mathcal{P}_{X_r}||\mathcal{P}_{\hat{X}_r}\right) + \alpha D\left(\mathcal{N}(0, \Sigma)||\mathcal{P}_{\hat{X}_f}\right) \right\}, \ \hat{X}_r = h_{\theta, \phi}\left(\mathcal{T}(X_r)\right), \ \hat{X}_f = h_{\theta, \phi}\left(\mathcal{T}(X_f)\right) \tag{6}$$

This way, we avoid the problem of the infinity value of KL-divergence in Eq. (4). We note that, for previous unlearning approaches for *classification* tasks, it's natural and straightforward to directly compute the KL-divergence for final outputs since the outputs are exactly single-variable discrete distributions after the SoftMax function (Zhang et al., 2023a;b; Kurmanji et al., 2023). Nevertheless, for image generation tasks, directly computing the KL divergence between high-dimensional output images is typically intractable, excluding the special case of diffusion models. To address this problem, we next convert the KL divergence into a more efficient $L_2$ loss which is generally applicable to diverse I2I generative models.

### 3.3 PROPOSED APPROACH

Directly connecting the KL-Divergence with the $L_2$ loss is difficult. Instead, we use Mutual Information (MI) as a bridge to help with the analysis. As indicated in Eq. (6), we reach the minimal objective value when $\mathcal{P}_{\hat{X}_r} = \mathcal{P}_{X_r}$ and $\mathcal{P}_{\hat{X}_f} = \mathcal{N}(0, \Sigma)$. This optimum can also be achieved by maximizing the mutual information ($I$) between $X_r$ and $\hat{X}_r$ (or between $n \sim \mathcal{N}(0, \Sigma)$ and $\hat{X}_f$); that is:

$$\arg\max_{\theta, \phi} \left\{ I\left(X_r; \hat{X}_r\right) + \alpha I\left(n; \hat{X}_f\right) \right\}, \ n \sim \mathcal{N}(0, \Sigma), \ \hat{X}_r = h_{\theta, \phi}\left(\mathcal{T}(X_r)\right), \ \hat{X}_f = h_{\theta, \phi}\left(\mathcal{T}(X_f)\right) \tag{7}$$

We next link the MI with a more tractable $L_2$ loss in the representation space.

**Theorem 1** *Suppose the original model can do a perfect generation, i.e., $h_{\theta_0, \phi_0}(\mathcal{T}(X)) = X$. Assume the target model $h_{\theta, \phi}$ uses the same decoder as the original model $h_{\theta_0, \phi_0}$ (i.e., $D_\phi = D_{\phi_0}$), and the output of the encoders is normalized, i.e., $\|E_\theta(x)\|_2 = \|E_{\theta_0}(x)\|_2 = 1$. On the retain set, minimizing the $L_2$ loss between the output of the target model encoder $E_\theta$ and the original model encoder $E_{\theta_0}$ will increase the lower bound of mutual information:*

$$I(X_r; \hat{X}_r) \geq \log(K) - \mathbb{E}\left[\sum_{i=1}^{K} \frac{1}{K} \log\left(e^{\frac{\epsilon_i^2}{2} - 1} \sum_{j=1}^{K} e^{\epsilon_j + R_{ij}}\right)\right] \tag{8}$$

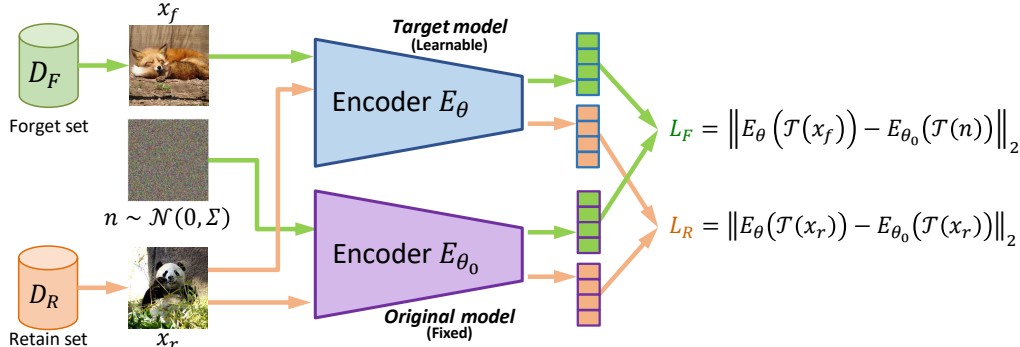

Figure 2: Overview of our approach. On $\mathcal{D}_F$, we minimize the $L_2$-loss between embedding vectors of the forget samples $x_f$ and embedding vectors of Gaussian noise $n$. On $\mathcal{D}_R$, we minimize the $L_2$-loss between the same image embedding vectors generated by target model encoder and the original model encoder.

where $\epsilon_i = \|E_\theta\left(\mathcal{T}(x_{r_i})\right) - E_{\theta_0}\left(\mathcal{T}(x_{r_i})\right)\|_2$ and $R_{ij} = E_{\theta_0}(\mathcal{T}(x_{r_i}))^T E_{\theta_0}(\mathcal{T}(x_{r_j}))$. $x_{r_i}$ are the data samples in the retain set. For the forget set, we have:

$$I(n; \hat{X}_f) \geq \log(K) - \mathbb{E}\left[\sum_{i=1}^{K} \frac{1}{K} \log\left(e^{\frac{\delta_i^2}{2}-1} \sum_{j=1}^{K} e^{\delta_j + F_{ij}}\right)\right], \quad n \sim \mathcal{N}(0, \Sigma) \quad (9)$$

where $\delta_i = \|E_\theta\left(\mathcal{T}(x_{f_i})\right) - E_{\theta_0}\left(\mathcal{T}(n_i)\right)\|_2$ and $F_{ij} = E_{\theta_0}(\mathcal{T}(n_i))^T E_{\theta_0}(\mathcal{T}(n_j))$. $x_{f_i}$ are the data samples in the forget set and $n_i \sim \mathcal{N}(0, \Sigma)$.

We remark that both $R_{ij}$ and $F_{ij}$ are determined by the original encoder $E_{\theta_0}$, thus are fixed values. As illustrated in Theorem 1, by directly reducing the $L_2$ loss ($\delta_i$ and $\epsilon_i$) between the target encoder and the original encoder, the Mutual Information (MI) increases, concurrently reducing the KL divergence between $\mathcal{P}_{X_r}$ and $\mathcal{P}_{\hat{X}_f}$ (or between $\mathcal{P}_{\hat{X}_f}$ and $\mathcal{N}$). Hence, in our approach, we sidestep the intractability of computing MI or KL divergence by directly minimizing the values of $\delta_i$ and $\epsilon_i$. Based on these insights, we next introduce our approach.

**Efficient Unlearning Approach.** Finally, as shown in Fig. 2, we propose our efficient unlearning approach for I2I generative models as follows:

$$A_F(h_{\theta_0,\phi_0}) \triangleq \arg\min_\theta \mathop{\mathbb{E}}_{x_{r_i}, x_{f_j}, n} \left\{ \left| E_\theta\left(\mathcal{T}(x_{r_i})\right) - E_{\theta_0}\left(\mathcal{T}(x_{r_i})\right) \right|_2 + \alpha \left| E_\theta\left(\mathcal{T}(x_{f_j})\right) - E_{\theta_0}\left(\mathcal{T}(n)\right) \right|_2 \right\}$$
$$x_{r_i} \in \mathcal{D}_R, x_{f_j} \in \mathcal{D}_F, n \sim \mathcal{N}(0, \Sigma) \quad (10)$$

We provide the details of our unlearning algorithm and corresponding pseudo code in Appendix C.4.

We note that our proposed approach only involves the encoders. Hence, it's more efficient than manipulating the entire model. Moreover, our approach is generally applicable to various I2I generative models with the encoder-decoder architecture (including the diffusion model, VQ-GAN, or MAE), although they typically use different optimization methods. We illustrate this generalizability in the experiments part.

## 4 EXPERIMENTAL RESULTS

We evaluate our proposed approach on three mainstream I2I generative models: (*i*) diffusion models (Saharia et al., 2022a), (*ii*) VQ-GAN (Li et al., 2023), and (*iii*) MAE (He et al., 2022).

### 4.1 EXPERIMENTAL SETUP

**Dataset&Task.** We verify our method on two mainstream large-scale datasets: (*i*) ImageNet-1k. Out of total 1K classes, we randomly select 100 classes as $\mathcal{D}_R$ and another 100 classes as $\mathcal{D}_F$. (*ii*)

Table 1: Results of cropping $8 \times 8$ patches at the center of the image, where each patch is $16 \times 16$ pixels. '↑' means higher is better and '↓' means lower is better. $R$ and $F$ account for the retain set and forget set, respectively. 'Proxy $\mathcal{D}_R$' means that we use the images from other classes as a substitute of the real retain set to do the unlearning (cf. Section 4.3).

| | Diffusion Models | | | | | | VQ-GAN | | | | | | MAE | | | | | |
| | FID | | IS | | CLIP | | FID | | IS | | CLIP | | FID | | IS | | CLIP | |
| | $R\downarrow$ | $F\uparrow$ | $R\uparrow$ | $F\downarrow$ | $R\uparrow$ | $F\downarrow$ | $R\downarrow$ | $F\uparrow$ | $R\uparrow$ | $F\downarrow$ | $R\uparrow$ | $F\downarrow$ | $R\downarrow$ | $F\uparrow$ | $R\uparrow$ | $F\downarrow$ | $R\uparrow$ | $F\downarrow$ |
|---|---|---|---|---|---|---|---|---|---|---|---|---|---|---|---|---|---|---|
| *Original model* | *12.2* | *14.6* | *19.3* | *23.1* | *0.88* | *0.89* | *14.4* | *14.4* | *19.4* | *20.6* | *0.75* | *0.77* | *56.7* | *84.1* | *23.0* | *17.4* | *0.73* | *0.71* |
| MAX LOSS | 34.1 | 45.7 | 12.8 | 17.1 | 0.77 | 0.76 | 16.9 | **115.2** | 17.4 | **11.0** | 0.73 | **0.55** | 75.8 | 112.6 | 19.4 | 15.2 | 0.69 | **0.65** |
| NOISY LABEL | 14.7 | 36.9 | 19.3 | 19.1 | 0.86 | 0.80 | 14.8 | 79.5 | 17.2 | 11.4 | **0.74** | 0.64 | 60.4 | 136.5 | 21.6 | 12.8 | 0.71 | 0.67 |
| RETAIN LABEL | 23.1 | 104.7 | 18.2 | 12.3 | 0.81 | **0.69** | 21.8 | 23.3 | 18.2 | 18.3 | 0.72 | 0.74 | 72.8 | 145.3 | 18.8 | 11.6 | 0.69 | 0.66 |
| RANDOM ENCODER | 15.3 | 30.6 | 18.7 | 19.4 | 0.86 | 0.81 | **14.7** | 72.8 | **18.6** | 14.1 | **0.74** | 0.64 | **58.1** | 146.4 | **22.3** | 12.8 | **0.72** | 0.67 |
| **Ours** | **13.4** | 107.9 | **19.4** | 10.3 | **0.87** | **0.69** | 15.0 | 83.4 | 18.3 | 11.6 | **0.74** | 0.60 | 59.9 | **153.0** | 21.8 | **11.0** | **0.72** | 0.67 |
| **Ours (Proxy $\mathcal{D}_R$)** | 17.9 | 75.5 | 18.2 | 12.3 | 0.83 | 0.74 | 17.6 | 69.7 | **18.6** | 14.0 | 0.73 | 0.63 | 61.1 | 133.8 | 21.0 | 12.3 | **0.72** | 0.68 |

Table 2: Results of cropping $4 \times 4$ patches at the center of the image, where each patch is $16 \times 16$ pixels. '↑' means higher is better and '↓' means lower is better. $R$ and $F$ account for the retain set and forget set, respectively. "Proxy $\mathcal{D}_R$" means that we use the images from other classes as a substitute of the real retain set to do the unlearning (cf. Section 4.3).

| | Diffusion Models | | | | | | VQ-GAN | | | | | | MAE | | | | | |
| | FID | | IS | | CLIP | | FID | | IS | | CLIP | | FID | | IS | | CLIP | |
| | $R\downarrow$ | $F\uparrow$ | $R\uparrow$ | $F\downarrow$ | $R\uparrow$ | $F\downarrow$ | $R\downarrow$ | $F\uparrow$ | $R\uparrow$ | $F\downarrow$ | $R\uparrow$ | $F\downarrow$ | $R\downarrow$ | $F\uparrow$ | $R\uparrow$ | $F\downarrow$ | $R\uparrow$ | $F\downarrow$ |
|---|---|---|---|---|---|---|---|---|---|---|---|---|---|---|---|---|---|---|
| *Original model* | *7.8* | *6.0* | *10.3* | *11.2* | *0.93* | *0.96* | *8.4* | *7.8* | *15.1* | *14.2* | *0.84* | *0.85* | *11.4* | *15.8* | *50.8* | *46.6* | *0.87* | *0.87* |
| MAX LOSS | 11.9 | 15.4 | 10.0 | 11.0 | 0.88 | 0.93 | 9.2 | **39.9** | 15.2 | **13.1** | 0.83 | **0.72** | 13.3 | 20.2 | **50.8** | 46.0 | **0.86** | 0.83 |
| NOISY LABEL | 19.6 | 18.5 | **10.4** | 10.6 | 0.87 | 0.91 | 8.7 | 21.3 | 15.2 | 14.1 | **0.84** | 0.80 | 12.2 | 44.3 | 50.0 | 35.4 | **0.86** | 0.82 |
| RETAIN LABEL | 8.5 | 35.1 | 10.3 | **10.5** | **0.93** | 0.89 | 11.0 | 10.3 | **15.4** | 14.2 | 0.83 | 0.84 | 15.3 | **47.5** | 47.6 | 34.9 | 0.85 | **0.81** |
| RANDOM ENCODER | 15.3 | 11.6 | 10.1 | 11.1 | 0.86 | 0.94 | **8.6** | 19.4 | 15.3 | 14.4 | **0.84** | 0.81 | **11.8** | 43.6 | 50.3 | 36.3 | **0.86** | 0.83 |
| **Ours** | **8.2** | 39.8 | 10.3 | 10.7 | **0.93** | 0.88 | **8.6** | 22.0 | 15.0 | 14.1 | **0.84** | 0.79 | 12.2 | 45.1 | 49.7 | **34.8** | **0.86** | 0.83 |
| **Ours (Proxy $\mathcal{D}_R$)** | 11.2 | 29.0 | 10.3 | 10.8 | 0.91 | 0.9 | 8.9 | 20.0 | **15.4** | 14.3 | **0.84** | 0.80 | 12.5 | 39.9 | 49.5 | 36.8 | **0.86** | 0.83 |

Places-365. From all 365 classes, we randomly select 50 classes as $\mathcal{D}_R$ and another 50 classes as $\mathcal{D}_F$. We test our method on image extension, uncropping, and reconstruction tasks. We report the results of center uncropping (i.e., inpainting) in the main paper. The results of other tasks are given in Appendix D and E.1.

**Baseline.** We first report the performance of the original model (i.e., before unlearning) as the reference. Since our approach is the first work that does the unlearning for I2I generative models, there are no previous baselines we can directly compare against. Therefore, we implement three different unlearning approaches that were designed for other tasks, and adapt them to I2I generative models, including (*i*) **MAX LOSS** maximizes the training loss w.r.t. the ground truth images on the forget set (Halimi et al., 2022; Gandikota et al., 2023; Warnecke et al., 2023); (*ii*) **NOISY LABEL** minimizes training loss by setting the Gaussian noise as the ground truth images for the forget set (Graves et al., 2021; Gandikota et al., 2023); (*iii*) **RETAIN LABEL** minimizes training loss by setting the retain samples as the ground truth for the forget set (Kong & Chaudhuri, 2023); (*iv*) **RANDOM ENCODER** directly minimizes the $L_2$ loss between the encoder's output on the forget set and a Gaussian noise (Tarun et al., 2023b). For all these baselines, we use the retain samples with some regularization to avoid hurting the performance on the retain set. For more details, please check Appendix C.6.

**Evaluation metrics.** We adopt three different types of metrics to compare our method against other baselines: (*i*) inception score (IS) of the generated images (Salimans et al., 2016), (*ii*) Fréchet inception distance (FID) against the real images (Heusel et al., 2017) and (*iii*) CLIP embedding distance between the generated images and the real images (Radford et al., 2021). IS assesses the quality of the generated images alone, while FID further measure the similarity between generated and real images. On the other hand, the CLIP embedding distance measures whether or not the generated images still capture similar semantics.

## 4.2 PERFORMANCE ANALYSIS AND VISUALIZATION

As shown in Table 1 and Table 2, compared to the original model, our approach has almost identical performance or only a slight degradation on the retain set. Meanwhile, there are significant perfor-

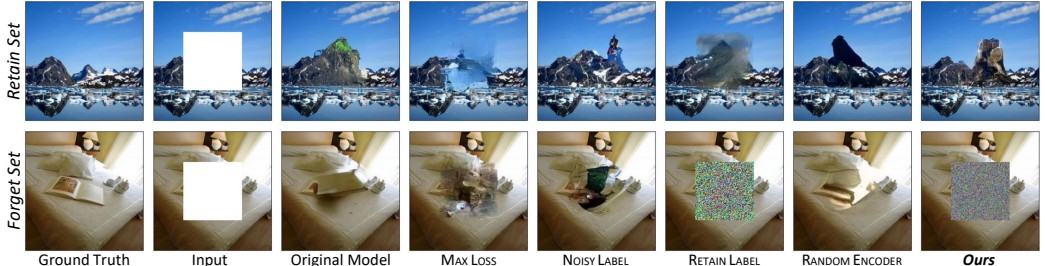

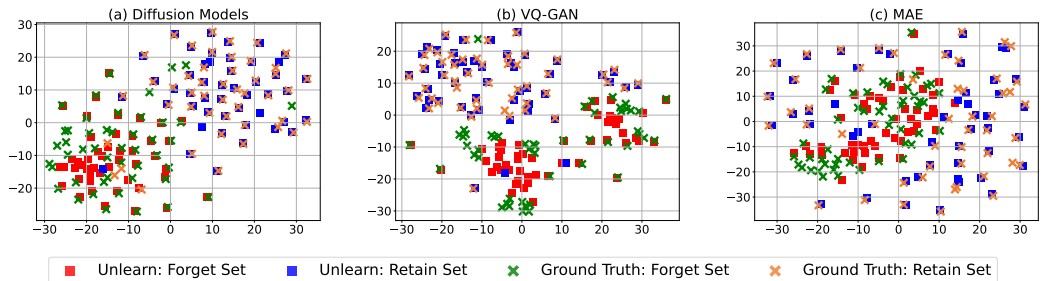

Figure 3: Results of cropping $8 \times 8$ patches at the center of the image on diffusion models, where each patch is $16 \times 16$ pixels. Our method has negligible-to-slight performance degradation on diverse I2I generative models and multiple generative tasks. (cf. Appendix D and E.1).

Figure 4: T-SNE analysis of the generated images by our approach and ground truth images. After unlearning, the generated retain samples are close to or overlapping with the ground truth (orange *vs.* blue), while most of generated forget images diverge far from the ground truth (green *vs.* red).

mance drops on the forget set across all these three models for all metrics. In contrast, none of these baselines generally works well. For example, RANDOM ENCODER achieves similar performance on VQ-GAN and MAE to our methods; however, it is much worse on diffusion models. Similarly, RETAIN LABEL works well for diffusion models, but cannot generalize to VQ-GAN and MAE. We also show some generated images in Fig. 3. As shown, our approach removes the information in the forget set while preserving the performance on the retain set.

**T-SNE analysis.** To further analyze why our approach works well, we conduct the T-SNE analysis. Using our unlearned model, we generate 50 images for both the retain and forget sets. We then compute the CLIP embedding vector of these images and their corresponding ground truth images. As shown in Fig. 4, after unlearning, the CLIP embedding vector on the retain set is close to or overlapping with the ground truth images, while most of generated images on the forget set diverge far from the ground truth.

These results verify that our method is generally applicable to mainstream I2I generative models and consistently achieves good results on all these models. We provide more results under various types of cropping in Appendix D and Appendix E.

### 4.3 ROBUSTNESS TO RETAIN SAMPLES AVAILABILITY

In machine unlearning, sometimes the real retain samples are not available due to data retention policies. To address this challenge, we evaluate our approach by using other classes of images as substitute to the real retain samples. On ImageNet-1K, since we already select 200 classes for forget and retain sets, we randomly select some images from the remaining 800 classes as the "proxy retain set" used in the unlearning process. Similarly, for Places-365, we randomly select some images from the remaining 265 classes as the "proxy retain set" used in the unlearning process. We also ensure these "proxy retain sets" have the same number of images as the forget set.

As shown in the last row in Table 1 and Table 2, our method works well even without the access to the real/original retain set. Compared to using the real/original retain set, there is only a slight performance drop. Hence, our approach is flexible and generally applicable without the dependency

Table 3: Ablation study of $\alpha$'s values. We test the performance of cropping $8 \times 8$ patches at the center of the image. As shown, $\alpha = 0.25$ achieves a good balance between the preserving the performance on retain set, while remove the information on forget sets across these two models.

| | | VQ-GAN | | | | | | | MAE | | | | | | |
|---|---|---|---|---|---|---|---|---|---|---|---|---|---|---|---|
| $\alpha$ | | 0.01 | 0.05 | 0.1 | 0.2 | **0.25** | 0.5 | 1 | 0.01 | 0.05 | 0.1 | 0.2 | **0.25** | 0.5 | 1 |
| FID | $R\downarrow$ | 90.8 | 91.6 | 92.0 | 91.7 | **92.7** | 92.2 | 94.7 | 113.6 | 113.2 | 113.9 | 116.7 | **115.9** | 116.3 | 116.7 |
| | $F\uparrow$ | 101.2 | 169.4 | 179.5 | 181.3 | **183.4** | 182.2 | 184.6 | 179.0 | 198.6 | 205.1 | 211.5 | **213.0** | 213.4 | 213.0 |
| IS | $R\uparrow$ | 12.5 | 12.8 | 12.5 | 12.4 | **12.2** | 12.0 | 12.6 | 13.3 | 13.3 | 13.4 | 13.5 | **13.2** | 13.3 | 12.9 |
| | $F\downarrow$ | 11.5 | 8.4 | 7.8 | 7.9 | **8.1** | 7.9 | 8.0 | 9.3 | 9.0 | 8.5 | 8.0 | **8.0** | 8.1 | 7.9 |
| CLIP | $R\uparrow$ | 0.65 | 0.65 | 0.65 | 0.65 | **0.65** | 0.65 | 0.64 | 0.81 | 0.81 | 0.81 | 0.80 | **0.80** | 0.80 | 0.80 |
| | $F\downarrow$ | 0.66 | 0.55 | 0.54 | 0.54 | **0.54** | 0.54 | 0.54 | 0.79 | 0.78 | 0.78 | 0.78 | **0.78** | 0.78 | 0.78 |

on the real retain samples. We provide the results with limited availability to the real retain samples in Appendix D.1.

## 4.4 ABLATION STUDY

For the ablation study, we test the results of cropping patches at the center of the image under various setups, where each patch is $16 \times 16$ pixels.

$\alpha$**'s value.** We vary the value of $\alpha$ in Eq. (10) to obtain multiple models and then evaluate their performance. As shown in Table 3, when $\alpha$ is 0.25, our approach achieves a good balance between the forget set and the retain set. Hence, we set $\alpha = 0.25$ as default value for our approach. We provide more ablation study in Appendix E.

## 5 CONCLUSIONS AND FINAL REMARKS

In this paper, we have formulated the machine unlearning problem for I2I generative models and derived an efficient algorithm that is applicable across various I2I generative models, including diffusion models, VQ-GAN, and MAE. Our method has shown negligible performance degradation on the retain set, while effectively removing the information from the forget set, on two large-scale datasets (ImageNet-1K and Places-365). Remarkably, our approach is still effective with limited or no real retain samples. To our best knowledge, we are the first to systematically explore machine unlearning for image completion generative models.

**Limitations.** First, our methods are mainly verified on I2I generative models. Second, our approach requires the access of original/real forget samples yet sometimes they are unavailable. Besides, for the simplicity of evaluation, we only test our approach on some mainstream computer vision datasets. Our approach has not been verified under a more practical/useful scenarios, e.g., remove the pornographic information for I2I generative models.

**Future directions.** We plan to explore applicability to other modality, especially for language/text generation and text-to-image generation. The dependency on the forget set is another challenge that enable flexibility in the unlearning for generative models. Finally, we also intend to develop some more practical benchmarks related to the control of generative contents and protect the data privacy and copyright.

## DISCLAIMER

**Ethics statement.** Machine unlearning for I2I generative models can be effectively exploited to avoid generate contents related user privacy and copyright. Moreover, unlearning for I2I models can avoid generating harmful contents, such as violence or pornography.

**Reproducibility statement.** All the datasets used in this paper are open dataset and are available to the public. Besides, our codes are primarily based on PyTorch (Paszke et al., 2019). We use several open source code base and model checkpoints to build our own approach (see Appendix C.1). Our approach can be implemented by obtaining the outputs of target model's encoders and the original model's encoders and then computing the $L_2$-loss between them. We provide more implementation details in Appendix C.

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

## A    INFINITELY MANY PROBABILITY WITH INFINITE KL-DIVERGENCE

In Section 3.2, we mention that there are infinitely many probability distributions that have infinity KL divergence with a given distribution. We provide the proof below:

**Proposition 1** *There are infinitely many probability distributions that have a positively infinite value of KL-divergence with any general discrete probability distribution $P(X)$ that is defined as follows:*

$$0 <= P(X = i) < 1, \quad \sum_{i=1}^{N} P(X = i) = 1, \quad i \in [N], \quad N \geq 2$$

*Proof.* Based on $P(X)$, we build another distribution $Q(X)$ as follows:

$$Q(X = i) = \begin{cases} \frac{P(X=i)}{1-P(X=j)-P(X=k)+\kappa}, & \text{if } i \neq j \text{ and } i \neq k \\ 0, & \text{if } i = j \\ \frac{\kappa}{1-P(X=j)-P(X=k)+\kappa}, & \text{if } i = k \end{cases} \quad j, k \in [N], \ \kappa > 0$$

where $j$ satisfies $P(X = j) > 0$ and $k$ sastifies $j \neq k$. Clearly, $0 <= Q(X = i) < 1$ and $\sum_{i=1}^{N} Q(X = i) = 1$. Therefore, $Q(X)$ is a valid probability distribution.

We next compute the KL divergence between $P$ and $Q$.

$$\begin{aligned} D(P||Q) &= \sum_{i=1}^{N} P(X=i) \log \frac{P(X=i)}{Q(X=i)} \\ &= P(X=j) \log \left( \frac{P(X=j)}{Q(X=j)} \right) + \sum_{i \in [N], i \neq j} P(X=i) \log \left( \frac{P(X=i)}{Q(X=i)} \right) \\ &= P(X=j) \log \left( \frac{P(X=j)}{0} \right) + \sum_{i \in [N], i \neq j} P(X=i) \log \left( \frac{P(X=i)}{Q(X=i)} \right) \\ &= +\infty \end{aligned}$$

We note that $\kappa$ can be any positive real number; hence, we can obtain infinitely many $Q(X)$ by varying the value of $\kappa$. Hence, there are infinitely many $Q$ that have a positively infinite value of KL-divergence with $P$. In general, one can replace the set $[N]$ with any discrete set. This completes our proof. $\qquad\square$

The proof for the continuous distribution is very similar to the discrete case as shown below.

**Proposition 2** *There are infinitely many probability distributions that have the a positively infinite value of KL-divergence with any general continuous probability distribution with the following probability density function (PDF) $p(x)$:*

$$p(x) \geq 0, \quad x \in \mathcal{S}, \quad \int_{x \in \mathcal{S}} p(x) dx = 1$$

*Proof.* Based on $p(x)$, we build another distribution with PDF $q(x)$ as follows:

$$q(x) = \begin{cases} \frac{p(x)}{1 - \int_{x \in \mathcal{S}_1} p(x)}, & \text{if } x \in \mathcal{S} \backslash (\mathcal{S}_1) \\ 0, & \text{if } x \in \mathcal{S}_1 \end{cases}, \quad \mathcal{S}_1 \subset \mathcal{S}$$

where $\mathcal{S}_1$ satisfies $0 < \int_{x \in \mathcal{S}_1} p(x) dx < 1$. Clearly, $q(x) \geq 0$ and $\int_{x \in \mathcal{S}} q(x) dx = 1$. Therefore, $q(x)$ is a valid probability density function.

We next compute the KL divergence between $p$ and $q$.

$$\begin{aligned} D(p||q) &= \int_{x \in \mathcal{S}} p(x) \log \left( \frac{p(x)}{q(x)} \right) dx \\ &= \int_{x \in \mathcal{S}_1} p(x) \log \left( \frac{p(x)}{q(x)} \right) dx + \int_{x \in \mathcal{S} \backslash \mathcal{S}_1} p(x) \log \left( \frac{p(x)}{q(x)} \right) dx \\ &= \int_{x \in \mathcal{S}_1} p(x) \log \left( \frac{p(x)}{0} \right) dx + \int_{x \in \mathcal{S} \backslash \mathcal{S}_1} p(x) \log \left( \frac{p(x)}{q(x)} \right) dx \\ &= +\infty \end{aligned}$$

We note that given a continuous distribution, there are infinitely many possible $\mathcal{S}_1$; hence, we can obtain infinitely many $q(x)$ by using different $\mathcal{S}_1$. Hence, there are infinitely many $q$ that have a positively infinite value of KL-divergence with $p$. This completes our proof. $\qquad\square$

## B   PROOF OF THEOREM 1

*Proof.* Directly computing mutual information (MI) between two random variables is not feasible. Fortunately, there are some variational MI bounds that are relatively easy to compute. A popular approach is to lower bound MI with InfoNCE (Belghazi et al., 2018; Wu et al., 2020; Poole et al., 2019; Kong et al., 2020). Given two random variables $X_1$ and $X_2$, MI is bounded by InfoNCE is defined as follows:

$$I(X_1; X_2) \geq \mathcal{I}^{NCE}(X_1; X_2) = \log(K) + \mathbb{E}\left[\sum_{i=1}^{K} \frac{1}{K} \log \frac{e^{g_1(x_{1_i})^T g_2(x_{2_i})}}{\sum_{j=1}^{K} e^{e^{g_1(x_{1_i})^T g_2(x_{2_j})}}}\right] \tag{B.11}$$

where the expectation is over $K$ independent samples from the joint distribution: $\Pi_j p(x_{1_j}, x_{2_j})$; $g_1$ and $g_2$ are two functions that map the random variables into representation vectors, e.g., the encoder network $E_\theta$ of I2I generative models.

Following the standard practice of InfoNCE, we use the inner product of the encoder output vectors as the critic function. Therefore, the InfoNCE between the ground truth images $X$ and reconstructed images $\hat{X}$ is written as follows:

$$\mathcal{I}^{NCE}(X; \hat{X}) = \log(K) + \mathbb{E}_{x,\hat{x}}\left[\frac{1}{K} \log \frac{e^{E_{\theta_0}(x_i)^T E_{\theta_0}(\hat{x}_i)}}{\sum_{j=1}^{K} e^{E_{\theta_0}(x_i)^T E_{\theta_0}(\hat{x}_j)}}\right] \tag{B.12}$$

Multiple works show that the encoder of the I2I generative models can be treated as the inverse function of the decoder (Wallace et al., 2023; Song et al., 2023; Xia et al., 2023; Kingma & Welling, 2019). In other words,

$$E_{\theta_0}(D_{\phi_0}(z)) = z \tag{B.13}$$

Given the assumption that the target model use the same decoder as the original model, we can express $X$ and $\hat{X}$ as follows:

$$x = D_{\phi_0}(E_{\theta_0}(\mathcal{T}(x))), \quad \hat{x} = D_{\phi_0}(E_\theta(\mathcal{T}(x))) \tag{B.14}$$

By using the above relationships in Eq. (B.13) and Eq. (B.14), we have

$$\begin{aligned}
E_{\theta_0}(x)^T E_{\theta_0}(\hat{x}) &= E_{\theta_0}(D_{\phi_0}(E_{\theta_0}(\mathcal{T}(x))))^T E_{\theta_0}(D_{\phi_0}(E_\theta(\mathcal{T}(\hat{x})))) \\
&= E_{\theta_0}(\mathcal{T}(x_1))^T E_\theta(\mathcal{T}(x_2))
\end{aligned} \tag{B.15}$$

**Retain set** Recall the assumption that the output of the encoder is normalized, i.e., $\|E_\theta(x)\|_2 = 1$ and $\|E_{\theta_0}(x)\|_2 = 1$. Therefore, we can rewrite Eq. (B.15) for the ground truth retain samples $\{x_{r_i}\}$ and their reconstructions $\{\hat{x}_{r_i}\}$ and as follows:

$$\begin{aligned}
E_{\theta_0}(x_{r_i})^T E_{\theta_0}(\hat{x}_{r_i}) &= E_{\theta_0}(\mathcal{T}(x_{r_i}))^T E_\theta(\mathcal{T}(x_{r_i})) \\
&= \frac{1}{2}\|E_{\theta_0}(\mathcal{T}(x_{r_i}))\|_2^2 + \frac{1}{2}\|E_\theta(\mathcal{T}(x_{r_i}))\|_2^2 \\
&\quad - \frac{1}{2}\|E_{\theta_0}(\mathcal{T}(x_{r_i})) - E_\theta(\mathcal{T}(x_{r_i}))\|_2^2 \\
&= \frac{1}{2}\left(2 - \|E_{\theta_0}(\mathcal{T}(x_{r_i})) - E_\theta(\mathcal{T}(x_{r_i}))\|_2^2\right) \\
&= \frac{1}{2}\left(2 - \epsilon_i^2\right) = 1 - \frac{\epsilon_i^2}{2}
\end{aligned} \tag{B.16}$$

where $\epsilon_i = \|E_{\theta_0}(\mathcal{T}(x_{r_i})) - E_\theta(\mathcal{T}(x_{r_i}))\|_2$ is the $L_2$ loss between the representation of target model encoder $E_\theta$ and original model encoder $E_{\theta_0}$. We then bound the $E_{\theta_0}(x_{r_i})^T E_{\theta_0}(\hat{x}_{r_j})$ as follows:

$$\begin{aligned}
E_{\theta_0}(x_{r_i})^T E_{\theta_0}(\hat{x}_{r_j}) &= E_{\theta_0}(\mathcal{T}(x_{r_i}))^T E_\theta(\mathcal{T}(x_{r_j})) \\
&= E_{\theta_0}(\mathcal{T}(x_{r_i}))^T \left(E_\theta(\mathcal{T}(x_{r_j})) - E_{\theta_0}(\mathcal{T}(x_{r_j}))\right) \\
&\quad + E_{\theta_0}(\mathcal{T}(x_{r_i}))^T E_{\theta_0}(\mathcal{T}(x_{r_j})) \\
&= E_{\theta_0}(\mathcal{T}(x_{r_i}))^T \left(E_\theta(\mathcal{T}(x_{r_j})) - E_{\theta_0}(\mathcal{T}(x_{r_j}))\right) + R_{ij} \\
&\leq \|E_{\theta_0}(\mathcal{T}(x_{r_i}))\|_2 * \|E_\theta(\mathcal{T}(x_{r_j})) - E_{\theta_0}\mathcal{T}(x_{r_j})\|_2 + R_{ij} \\
&= 1 * \epsilon_j + R_{ij} \\
&= \epsilon_j + R_{ij}
\end{aligned} \tag{B.17}$$

where $R_{ij} = E_{\theta_0} \left( \mathcal{T} \left( x_{r_i} \right) \right)^T E_{\theta_0} \left( \mathcal{T} \left( x_{r_j} \right) \right)$ and the '$\leq$' comes from the Cauchy–Schwarz inequality. The above bound is tight if $\{ \epsilon_i = 0, i \in [K] \}$. By combining the Eq. (B.12), Eq. (B.16), and Eq. (B.17), we can link the InfoNCE loss with the $L_2$ loss:

$$
\begin{aligned}
\mathcal{I}^{NCE}(X_r; \hat{X}_r) &= \log\left(K\right) + \mathbb{E}\left[ \frac{1}{K} \log \frac{e^{E_{\theta_0}(x_i)^T E_{\theta_0}(\hat{x}_i)}}{\sum_{j=1}^{K} e^{E_{\theta_0}(x_i)^T E_{\theta_0}(\hat{x}_j)}} \right] \\
&= \log\left(K\right) + \mathbb{E}\left[ \frac{1}{K} \log \frac{e^{1 - \frac{\epsilon_i^2}{2}}}{\sum_{j=1}^{K} e^{E_{\theta_0}(x_i)^T E_{\theta_0}(\hat{x}_j)}} \right] \\
&\geq \log\left(K\right) + \mathbb{E}\left[ \frac{1}{K} \log \frac{e^{1 - \frac{\epsilon_i^2}{2}}}{\sum_{j=1}^{K} e^{\epsilon_j + R_{ij}}} \right] \\
&= \log\left(K\right) - \mathbb{E}\left[ \sum_{i=1}^{K} \frac{1}{K} \log \left( e^{\frac{\epsilon_i^2}{2} - 1} \sum_{j=1}^{K} e^{\epsilon_j + R_{ij}} \right) \right]
\end{aligned}
\tag{B.18}
$$

By combining Eq. (B.11), we obtain the results on the retain set:

$$
I(X_r; \hat{X}_r) \geq \log\left(K\right) - \mathbb{E}\left[ \sum_{i=1}^{K} \frac{1}{K} \log \left( e^{\frac{\epsilon_i^2}{2} - 1} \sum_{j=1}^{K} e^{\epsilon_j + R_{ij}} \right) \right]
\tag{B.19}
$$

**Forget set** The proof on the forget set is very similar to the retain set. By adapting Eq. (B.16) and Eq. (B.17) for the forget set, we first calculate the inner product of the embedding vector between $n$ and $x_f$:

$$
\begin{aligned}
E_{\theta_0}\left(n_i\right)^T E_{\theta_0}\left(\hat{x}_{f_i}\right) &= E_{\theta_0}\left(\mathcal{T}\left(n_i\right)\right)^T E_{\theta}\left(\mathcal{T}\left(x_{f_i}\right)\right) \\
&= \frac{1}{2}\left( \|E_{\theta_0}\left(\mathcal{T}\left(n_i\right)\right)\|_2^2 + \|E_{\theta}\left(\mathcal{T}\left(x_{f_i}\right)\right)\|_2^2 - \|E_{\theta_0}\left(\mathcal{T}\left(n_i\right)\right) - E_{\theta}\left(\mathcal{T}\left(x_{f_i}\right)\right)\|_2^2 \right) \\
&= \frac{1}{2}\left( 2 - \|E_{\theta_0}\left(\mathcal{T}\left(n_i\right)\right) - E_{\theta}\left(\mathcal{T}\left(x_{f_i}\right)\right)\|_2^2 \right) \\
&= \frac{1}{2}\left( 2 - \epsilon_i^2 \right) = 1 - \frac{\delta_i^2}{2}
\end{aligned}
\tag{B.20}
$$

and

$$
\begin{aligned}
E_{\theta_0}\left(n_i\right)^T E_{\theta_0}\left(\hat{x}_{f_j}\right) &= E_{\theta_0}\left(\mathcal{T}\left(n_i\right)\right)^T E_{\theta}\left(\mathcal{T}\left(x_{f_j}\right)\right) \\
&= E_{\theta_0}\left(\mathcal{T}\left(n_i\right)\right)^T \left( E_{\theta}\left(\mathcal{T}\left(x_{f_j}\right)\right) - E_{\theta_0}\left(\mathcal{T}\left(n_j\right)\right) \right) \\
&\quad + E_{\theta_0}\left(\mathcal{T}\left(n_i\right)\right)^T E_{\theta_0}\left(\mathcal{T}\left(n_j\right)\right) \\
&= \|E_{\theta_0}\left(\mathcal{T}\left(n_i\right)\right)\|_2 * \|E_{\theta}\left(\mathcal{T}\left(x_{f_j}\right)\right) - E_{\theta_0}\mathcal{T}\left(n_j\right)\|_2 + F_{ij} \\
&= 1 * \delta_j + F_{ij} \\
&= \delta_j + F_{ij}
\end{aligned}
\tag{B.21}
$$

where $\delta_i = \|E_{\theta_0}\left(\mathcal{T}\left(n_i\right)\right) - E_{\theta}\left(\mathcal{T}\left(x_{f_i}\right)\right)\|_2$ and $F_{ij} = E_{\theta_0}\left(\mathcal{T}\left(n_i\right)\right)^T E_{\theta_0}\left(\mathcal{T}\left(n_j\right)\right)$. $x_{f_i}$ are the data samples in the forget set and $n_i \sim \mathcal{N}(0, \Sigma)$. Combining the above two equation with Eq. (B.12):

$$
\begin{aligned}
\mathcal{I}^{NCE}(n; \hat{X}_f) &= \log\left(K\right) + \mathbb{E}\left[ \frac{1}{K} \log \frac{e^{E_{\theta_0}(n_i)^T E_{\theta_0}(\hat{x}_{f_i})}}{\sum_{j=1}^{K} e^{E_{\theta_0}(n_i)^T E_{\theta_0}(\hat{x}_{f_j})}} \right] \\
&= \log\left(K\right) + \mathbb{E}\left[ \frac{1}{K} \log \frac{e^{1 - \frac{\delta_i^2}{2}}}{\sum_{j=1}^{K} e^{E_{\theta_0}(n_i)^T E_{\theta_0}(\hat{x}_{f_j})}} \right] \\
&\geq \log\left(K\right) + \mathbb{E}\left[ \frac{1}{K} \log \frac{e^{1 - \frac{\delta_i^2}{2}}}{\sum_{j=1}^{K} e^{\delta_j + F_{ij}}} \right] \\
&= \log\left(K\right) - \mathbb{E}\left[ \sum_{i=1}^{K} \frac{1}{K} \log \left( e^{\frac{\delta_i^2}{2} - 1} \sum_{j=1}^{K} e^{\delta_j + F_{ij}} \right) \right]
\end{aligned}
\tag{B.22}
$$

By combining the above equation with Eq. (B.11), we obtain the results for the forget set:

$$I(n; \hat{X}_f) \geq \log(K) - \mathbb{E}\left[\sum_{i=1}^{K} \frac{1}{K} \log\left(e^{\frac{\delta_i^2}{2}-1} \sum_{j=1}^{K} e^{\delta_j + F_{ij}}\right)\right] \tag{B.23}$$

This completes our proof. $\qquad\square$

## C  IMPLEMENTATION DETAILS

### C.1  DATASETS AND CODE BASE

**Diffusion Models.** We verify our approach on a diffusion model that is trained on entire Places-365 training dataset (Saharia et al., 2022a; Zhou et al., 2017). We randomly select 50 classes out of 365 classes as the retain set and another 50 classes as the forget set. For each class, we select 5000 images from the Places-365 training set; we then combine them together as the training set for unlearning. By using the approach defined in Eq. (10), we obtain the target model. We then evaluate the obtained model on both forget set and retain set, with 100 images per class from the Places-365 validation set. Hence, we have 5,000 validation images for both the retains and forget sets. Since Saharia et al. did not release the code, our code is implemented based on an open source re-implementation (GitHub). Our experiments are using their provided model checkpoint on Places-365 dataset.

**VQ-GAN** We evaluate our approach on a VQ-GAN model that is trained on entire ImageNet-1K training dataset (Li et al., 2023; Deng et al., 2009). We randomly select 100 classes out of 1000 classes as the retain set and another 100 classes as the forget set. We select 100 images per class from the training set as the training set for unlearning. We then apply our proposed approach to the original model and obtain the target model. For the main results reported in Tabel 1 and Tabel 2, we evaluate the obtained model on both forget set and retain set, with 50 images per class from the ImageNet-validation set; hence, we have 5,000 validation images for both the retains set and forget set. For the other results (in ablation study), we use a smaller version validation set, with 5 images per class from the ImageNet-validation set; hence, we have 500 validation images for both the retains set and forget set. Our code is implemented based on the offically released code of Li et al. (2023) (GitHub). Our experiments are using their provided ViT-Base model checkpoints.

**MAE** We evaluate our approach on a MAE model and an MAE model that is trained on the entire ImageNet-1K training dataset (He et al., 2022). The dataset setup is exactly the same as VQ-GAN (check the upper paragraph). Our code is implemented based on the offically released code of He et al. (2022) (GitHub). Our experiments are using their provided ViT-Base model checkpoints.

### C.2  EVALUATION METRICS

**Inception score (IS)** For ImageNet-1K, we directly use the Inception-v3 model checkpoint from torchvision library to compute the IS. For Places-365, we used the ResNet-50 model checkpoint from the official release to compute IS (Zhou et al., 2017).

**Fréchet inception distance (FID)** For both ImageNet-1K and Places-365, we directly use the backbone network of the Inception-v3 model checkpoint from Torchvision library to compute the FID.

**CLIP** We use CLIP with ViT-H-14 as the backbone to generate the embedding vectors of each reconstructed image from the original model and unlearned model (Radford et al., 2021). Then we compute the cosine similarity of these embedding vectors among $\mathcal{D}_F$ and $\mathcal{D}_R$ separately.

### C.3  TRAINING HYPER-PARAMETERS

**Patch Size** For all of the models we evaluate in this paper (including diffusion models, VQ-GAN and MAE), the networks are vision transformer (Dosovitskiy et al., 2021) based architecture. We set the size of each patch as $16 \times 16$ pixels for all experiments. For example, cropping $8 \times 8$ patches means removing $128 \times 128$ pixels. We set $\alpha = 0.25$ (cf. Eq. (10)).

**Diffusion Models** We set the learning rate as $10^{-5}$ with no weight decay. We use the Adam as the optimizer and conduct the unlearning for 3 epochs. We set the input resolution as 256 and set the batch size as 8 per GPU. Overall, it takes 1.5 hours on a 8 NVIDIA A10G server. We set $\alpha = 0.25$ (cf. Eq. (10)).

**VQ-GAN&MAE** We set the learning rate as $10^{-4}$ with no weight decay. We use the AdamW as the optimizer with $\beta = (0.9, 0.95)$ and conduct the unlearning for 5 epochs. We set the input resolution as 256 for VQ-GAN and 224 for MAE. We set the batch size as 16 per GPU. Overall, it takes one hour on a 4 NVIDIA A10G server. We set $\alpha = 0.25$ (cf. Eq. (10)).



(a) MNIST      (b) CIFAR-10      (c) CIFAR-100

Figure C.5: Covariance matrix of three commonly datasets. For CIFAR10/100, we convert the images into gray-scale images. We take the absolute value of the covariance matrix for better illustration.

---

**Algorithm 1** Pseudo Code of Our Unlearning Algorithm

1: **Inputs:**
        Orginal model $h_{\theta_0,\phi_0} = D_{\phi_0} \circ E_{\theta_0}$
        Retain set $\mathcal{D}_R$, Forget set $\mathcal{D}_F$
        Coefficient $\alpha$, learning rate $\zeta$, and #Epochs E
2: **Outputs:**
        Target model $h_{\theta,\phi} = D_\phi \circ E_\theta$
3: **Initialize:**
        Copy $h_{\theta,\phi}$ to $h_{\theta,\phi}$, i.e., $h_{\theta,\phi} \Leftarrow h_{\theta_0,\phi_0}$
4: **for** e = 1 to E **do**
5:       Sample $\{x_r\}$ from $\mathcal{D}_R$
6:       Sample $\{x_f\}$ from $\mathcal{D}_F$
7:       Sample $\{n\}$ from $\mathcal{N}(0,\Sigma)$
8:       **Ensure:** $|\{x_r\}| = |\{x_f\}|$, i.e., make retain samples and forget samples balanced
9:       Compute loss: $l = \|E_\theta(\mathcal{T}(x_r)) - E_{\theta_0}(\mathcal{T}(x_r))\|_2 + \alpha \|E_\theta(\mathcal{T}(x_f)) - E_{\theta_0}(\mathcal{T}(n))\|_2$
10:     Update the parameters of the target encoder $E_\theta$: $\theta \Leftarrow \theta - \zeta \nabla_\theta l$
11: **end for**

---

## C.4   Our Unlearning Algorithm

**Selection of $\Sigma$ in Eq. (10)**    To conduct our unlearning algorithm, we need to compute the $\Sigma$ in Eq. (10), where $\Sigma$ is the the covariance matrix for the distribution of training images.

Ideally, we should use the exact $\Sigma$ of the images measured on the forget set. However, there are some computational barriers to using the exact $\Sigma$ for a high-resolution image dataset. Specifically, consider a commonly used 256×256 resolution for image generation tasks, the distribution of the generated images will have $256 \times 256 \times 3 \approx 2 \times 10^5$ dimensions. The size of the covariance matrix $\Sigma$ for such a high-dimensional distribution is around $(2 \times 10^5)^2 = 4 \times 10^{10}$, which requires around 144GB memory if stored in float precision thus is not practical.

Consequently, to address the computational barrier of $\Sigma$, we use some approximated methods derived from some empirical observations on some small datasets. Specifically, we compute the exact $\Sigma$ for some small-scale image dataset, including MNIST and CIFAR10/100.

- Off-diagonal elements: To find some empirical inspirations, we compute the covariance matrix for three commonly dataset, MNIST, CIFAR10 and CIFAR100. As shown in Fig. C.5, most of the off-diagonal elements are very close to '0'. Hence, in our experiments, we set the off-diagonal elements of $\Sigma$ to '0'

- Diagonal elements: Since the images are normalized (i.e., subtract the mean value and divided by the standard deviation), we set the diagonal elements of $\Sigma$ as '1', i.e., $\Sigma(i,i) = 1$.

Therefore, we use the identity matrix as the approximation of exact $\Sigma$:

$$\Sigma = \boldsymbol{I}$$

where $\boldsymbol{I}$ is the identity matrix. We set $\Sigma = \boldsymbol{I}$ as the default setup for our approach and baseline methods.

In short, using $I$ to approximate $\Sigma$ is a practical approximation alternative due to the extremely high computational costs of $\Sigma$ for high-resolution images. For future work, given our theoretical analysis, we believe that

our approach can achieve better results (lower image quality) on forget set if we can find a way to use the exact $\Sigma$. Hence, we plan to explore the potential to reduce the computation of exact $\Sigma$ with low-rank approximation thus enabling the use of more accurate data-driven $\Sigma$.

**Pseudo code**   Based on our unlearning approach defined in Section 3.3, we implement the unlearning algorithm for I2I generative models. We provide the pseudo code in Algorithm 1. As shown, for each batch, we sample the same number of retain samples and forge samples. We then compute the loss by using Eq. (10) and update the parameters of the target model's encoder.

## C.5   LOSS FUNCTION OF OUR APPROACH

As shown in Eq. (10), we input the Gaussian noise to the original model as the reference for the forget set. For VQ-GAN and MAE, we can directly use Eq. (10). For diffusion models, in principle, we still replace the forget sample with Gaussian noise as the input for the original encoder. We next discuss the details for the diffusion models.

We first write the loss function for the normal training of diffusion models first:

$$\mathbb{E}_x \mathbb{E}_{\boldsymbol{\gamma} \sim \mathcal{N}(0,I)} \mathbb{E}_\eta \left\| D_\phi \circ E_\theta \left( \mathcal{T}(x),\ \underbrace{\sqrt{\eta}x + \sqrt{1-\eta}\,\boldsymbol{\gamma}}_{\tilde{x}},\ \eta \right) - \boldsymbol{\gamma} \right\|_p \tag{C.24}$$

where $x$ is the ground truth image; $\mathcal{T}(x)$ is the transformed images, e.g., a cropped image for image uncropping tasks and and low-resolution image for image super-resolution. $\eta$ is the forward process variance coefficient. For more details, please refer to the Saharia et al. (2022a).

Now, we introduce the unlearning optimization function for diffusion models:

$$\arg\min_\theta \mathbb{E}_{x_r,x_f,n} \mathbb{E}_{\boldsymbol{\gamma} \sim \mathcal{N}(0,I)} \mathbb{E}_\eta \Big\{\ \left\| E_\theta\left(\mathcal{T}(x_r),\ \tilde{x_r},\ \eta\right) - E_{\theta_0}\left(\mathcal{T}(x_r),\ \tilde{x_r},\ \eta\right) \right\|_2$$
$$+\alpha\ \left\| E_\theta\left(\mathcal{T}(x_f),\ \tilde{x_f},\ \eta\right) - E_{\theta_0}\left(\mathcal{T}(x_f),\ n,\ \eta\right) \right\|_2 \Big\}, \tag{C.25}$$
$$x_r \in \mathcal{D}_R, x_f \in \mathcal{D}_F, n \sim \mathcal{N}(0,\Sigma)$$

where $\tilde{x_r} = \sqrt{\eta}x_r + \sqrt{1-\eta}\,\boldsymbol{\gamma}$ and $\tilde{x_f} = \sqrt{\eta}x_f + \sqrt{1-\eta}\,\boldsymbol{\gamma}$. Essentially, we replace the $\tilde{x_f}$ with the Gaussian noise $n$ as the input for the original encoder. Note that, Saharia et al. adopt an U-Net as the network architecture; thus the encoder has multiple-stage outputs. We flatten these multiple outputs into vectors and then combine them as a single representation vector .

We remark the equation for diffusion models looks slightly different from VQ-GAN and MAE, but they are actually following the same principle defined in Eq. (10); that is, we replace the real forget samples with the Gaussian noise as the input for the original encoder.

## C.6   BASELINES

We use the exact same setup as introduced in the main paper and Appendix C.3. We provide the loss function for different baseline methods below. For the baselines, the unlearning is achieved by minimizing the loss values.

### C.6.1   MAX LOSS

MAX LOSS maximizes the training loss w.r.t. the ground truth images on the forget set. Hence, we use the original loss function for both retains set and forget set, but assign a negative coefficient for the forget set. We do not modify the loss for the retains set.

- Diffusion Models

$$\arg\min_{\theta,\phi} \mathbb{E}_{x_r,x_f} \mathbb{E}_{\boldsymbol{\gamma} \sim \mathcal{N}(0,I)} \mathbb{E}_\eta \Big\{ \left\| D_\phi \circ E_\theta\left(\mathcal{T}(x_r), \tilde{x_r}, \eta\right) - \boldsymbol{\gamma} \right\|_2 - \alpha\ \left\| D_\phi \circ E_\theta\left(\mathcal{T}(x_f),\ \tilde{x_f}, \eta\right) - \boldsymbol{\gamma} \right\|_2 \Big\} \tag{C.26}$$

where $\tilde{x_r} = \sqrt{\eta}x_r + \sqrt{1-\eta}\,\boldsymbol{\gamma}$ and $\tilde{x_f} = \sqrt{\eta}x_f + \sqrt{1-\eta}\,\boldsymbol{\gamma}$. $\mathcal{T}(\cdot)$ is the transformed function, e.g., a cropped image for image uncropping tasks.

- VQ-GAN

$$\underset{\theta,\phi}{\arg\max} \, \mathbb{E}_{x_r,x_f} \left\{ C_{\mathcal{J}}\left(D_\phi \circ E_\theta\left(\mathcal{T}(x_r)\right), x_r\right) - \alpha C_{\mathcal{J}}\left(D_\phi \circ E_\theta\left(\mathcal{T}(x_f)\right), x_f\right) \right\} \tag{C.27}$$

where $\mathcal{J}$ is a discriminator network used to predict whether the images are real images or generated images; $C_{\mathcal{J}}$ is the cross entropy of the discriminator prediction. We also note that, during unlearning, the discriminator $\mathcal{J}$ is also updated in the normal training way. Please check more details in Li et al. (2023).

- MAE

$$\underset{\theta,\phi}{\arg\min} \, \mathbb{E}_{x_r,x_f} \left\{ \left\| D_\phi \circ E_\theta\left(\mathcal{T}(x_r)\right) - x_r \right\|_2 - \alpha \, \left\| D_\phi \circ E_\theta\left(\mathcal{T}(x_f)\right) - x_f \right\|_2 \right\} \tag{C.28}$$

$\mathcal{T}(\cdot)$ is the transformed function, e.g., a cropped image for image uncropping tasks.

### C.6.2 RETAIN LABEL

RETAIN LABEL minimizes training loss by setting the retain samples as the ground truth for the forget set.

- Diffusion Models

$$\underset{\theta,\phi}{\arg\min} \, \mathbb{E}_{x_r,x_f} \mathbb{E}_{\gamma \sim \mathcal{N}(0,I)} \mathbb{E}_\eta \left\{ \left\| D_\phi \circ E_\theta\left(\mathcal{T}(x_r), \tilde{x}_r, \eta\right) - \gamma \right\|_2 + \alpha \, \left\| D_\phi \circ E_\theta\left(\mathcal{T}(x_f), \, \tilde{x}_r, \eta\right) - \gamma \right\|_2 \right\} \tag{C.29}$$

where $\tilde{x}_r = \sqrt{\eta} x_r + \sqrt{1-\eta}\,\gamma$. As shown, the $\tilde{x}_f$ in Eq. (C.26) is replaced by the the retain samples $\tilde{x}_r$.

- VQ-GAN

$$\underset{\theta,\phi}{\arg\max} \, \mathbb{E}_{x_r,x_f} \left\{ C_{\mathcal{J}}\left(D_\phi \circ E_\theta\left(\mathcal{T}(x_r)\right), x_r\right) + \alpha C_{\mathcal{J}}\left(D_\phi \circ E_\theta\left(\mathcal{T}(x_f)\right), x_r\right) \right\} \tag{C.30}$$

where $\mathcal{J}$ is a discriminator network used to predict whether the images are real images or generated images; $C_{\mathcal{J}}$ is the cross entropy of the discriminator prediction. As shown, the reference $x_f$ in the second term of Eq. (C.27) is replaced by the retain sample $x_r$.

- MAE

$$\underset{\theta,\phi}{\arg\min} \, \mathbb{E}_{x_r,x_f} \left\{ \left\| D_\phi \circ E_\theta\left(\mathcal{T}(x_r)\right) - x_r \right\|_2 + \alpha \, \left\| D_\phi \circ E_\theta\left(\mathcal{T}(x_f)\right) - n \right\|_2 \right\} \tag{C.31}$$

As shown, the target $\tilde{x}_f$ in the second term of Eq. (C.28) is replaced by the retain sample $x_r$.

### C.6.3 NOISY LABEL

NOISY LABEL minimizes training loss by setting the Gaussian noise as the ground truth images for the forget set. Hence we directly use the original loss function by replacing the ground truth images with standard Gaussian noise for the forget set. We do not modify the loss for the retains set.

- Diffusion Models

$$\underset{\theta,\phi}{\arg\min} \, \mathbb{E}_{x_r,x_f} \mathbb{E}_{\gamma \sim \mathcal{N}(0,I)} \mathbb{E}_\eta \left\{ \left\| D_\phi \circ E_\theta\left(\mathcal{T}(x_r), \tilde{x}_r, \eta\right) - \gamma \right\|_2 + \alpha \, \left\| D_\phi \circ E_\theta\left(\mathcal{T}(x_f), \, \eta, \eta\right) - \gamma \right\|_2 \right\} \tag{C.32}$$

where $\tilde{x}_r = \sqrt{\eta} x_r + \sqrt{1-\eta}\,\gamma$. As shown, the $\tilde{x}_f$ in Eq. (C.26) is replaced by the Gaussian noise $\eta$. $\mathcal{T}(\cdot)$ is the transformed function, e.g., a cropped image for image uncropping tasks.

- VQ-GAN

$$\underset{\theta,\phi}{\arg\max} \, \mathbb{E}_{x_r,x_f} \left\{ C_{\mathcal{J}}\left(D_\phi \circ E_\theta\left(\mathcal{T}(x_r)\right), x_r\right) + \alpha C_{\mathcal{J}}\left(D_\phi \circ E_\theta\left(\mathcal{T}(x_f)\right), n\right) \right\} \tag{C.33}$$

where $\mathcal{J}$ is a discriminator network used to predict whether the images are real images or generated images; $C_{\mathcal{J}}$ is the cross entropy of the discriminator's prediction. As shown, the reference $x_f$ in the second term of Eq. (C.27) is replaced by the Gaussian noise $n$.

- MAE

$$\underset{\theta,\phi}{\arg\min} \, \mathbb{E}_{x_r,x_f} \left\{ \left\| D_\phi \circ E_\theta\left(\mathcal{T}(x_r)\right) - x_r \right\|_2 + \alpha \, \left\| D_\phi \circ E_\theta\left(\mathcal{T}(x_f)\right) - n \right\|_2 \right\} \tag{C.34}$$

As shown, the target $\tilde{x}_f$ in the second term of Eq. (C.28) is replaced by the Gaussian noise $n$.

### C.6.4 RANDOM ENCODER

RANDOM ENCODER minimizes the $L_2$ loss between the Gaussian noise and the representation vector of the encoder for the forget set.

- Diffusion Models

$$\arg\min_{\theta} \mathbb{E}_{x_r, x_f} \mathbb{E}_{\gamma \sim \mathcal{N}(0,I)} \mathbb{E}_{\eta} \Bigg\{ \left\| E_{\theta}\left(\mathcal{T}(x_r),\ \tilde{x}_r,\ \eta\right) - E_{\theta_0}\left(\mathcal{T}(x_r),\ \tilde{x}_r,\ \eta\right) \right\|_2 \\ + \alpha \left\| E_{\theta}\left(\mathcal{T}(x_f),\ \tilde{x}_f,\ \eta\right) - \eta \right\|_2 \Bigg\} \tag{C.35}$$

where $\tilde{x}_r = \sqrt{\eta}x_r + \sqrt{1-\eta}\,\gamma$. As shown, the target for the forget set is directly the Gaussian noise instead of its embedding vector.

- VQ-GAN&MAE

$$\arg\min_{\theta} \mathbb{E}_{x_r, x_f} \Bigg\{ \left\| E_{\theta}\left(\mathcal{T}(x_r)\right) - E_{\theta_0}\left(\mathcal{T}(x_r)\right) \right\|_2 + \alpha \left\| E_{\theta}\left(\mathcal{T}(x_f)\right) - n \right\|_2 \Bigg\} \tag{C.36}$$

## D SUPPLEMENTARY RESULTS

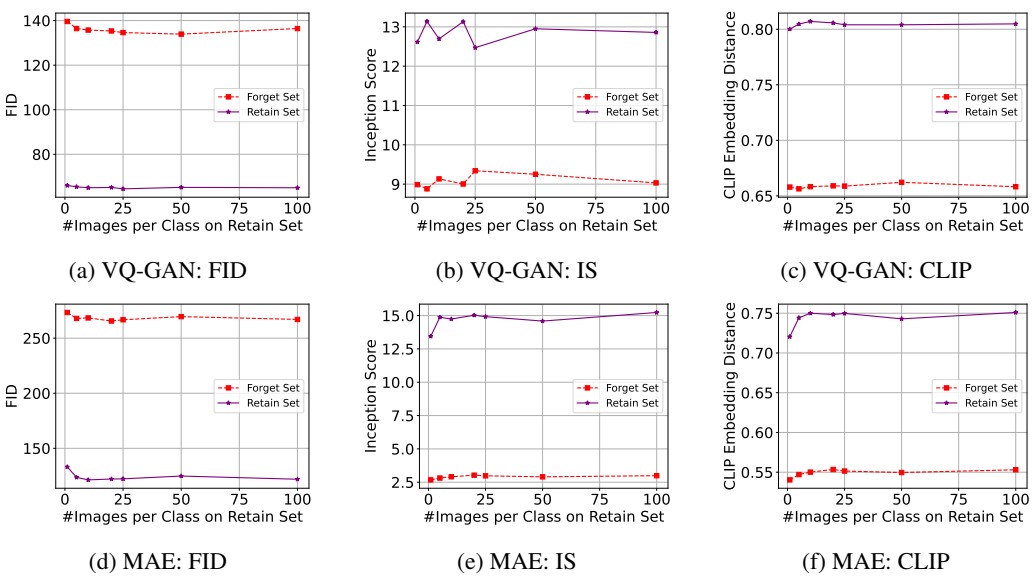

| | | |
|---|---|---|
| (a) VQ-GAN: FID | (b) VQ-GAN: IS | (c) VQ-GAN: CLIP |
| (d) MAE: FID | (e) MAE: IS | (f) MAE: CLIP |

Figure D.6: The performance of our approach under limited availability of the retain sample. The "100 images per class" (right side of horizontal-axis) in these plots indicate a *full* retains set baseline. As shown, by gradually reducing the number of images for the retain set, the performance degradation is negligible. Even for the extreme case where we only one image per class, the performance degradation is also small.

### D.1 ROBUSTNESS TO RETAIN SAMPLES AVAILABILITY

In Table 1 and Table 2, we report the results under an extreme case where there is no real retain sample available. In this section, we relax the constraints by assuming the limited availability to the real retain samples. Specifically, on ImageNet-1K, we fix the forget set by using 100 images per class for the forget set; in total, we have 10K image for the forget set. As the baseline method, we have a retain set with 100 image per class as well (in total 10K); we call this a *full* retain set.

We next vary the number of images per class for the retain set within the range of $\{1, 5, 10, 20, 25, 50\}$ and compare the performance of the unlearned models under different values. To balance the number of forget sample and retain samples, we over sample the retain samples. For example, if we have 5 image per class for the retain set, we sample these images 20 times (since $100/5 = 20$).

As shown in Fig. D.6, compared to the full retain set (100 images per class), our approach is still effective with a slight performance degradation. Combining the results in Table 1 and Table 2, these results show the applicability and resilience of our approach in scenarios where actual the retain samples are limited or even unavailable.

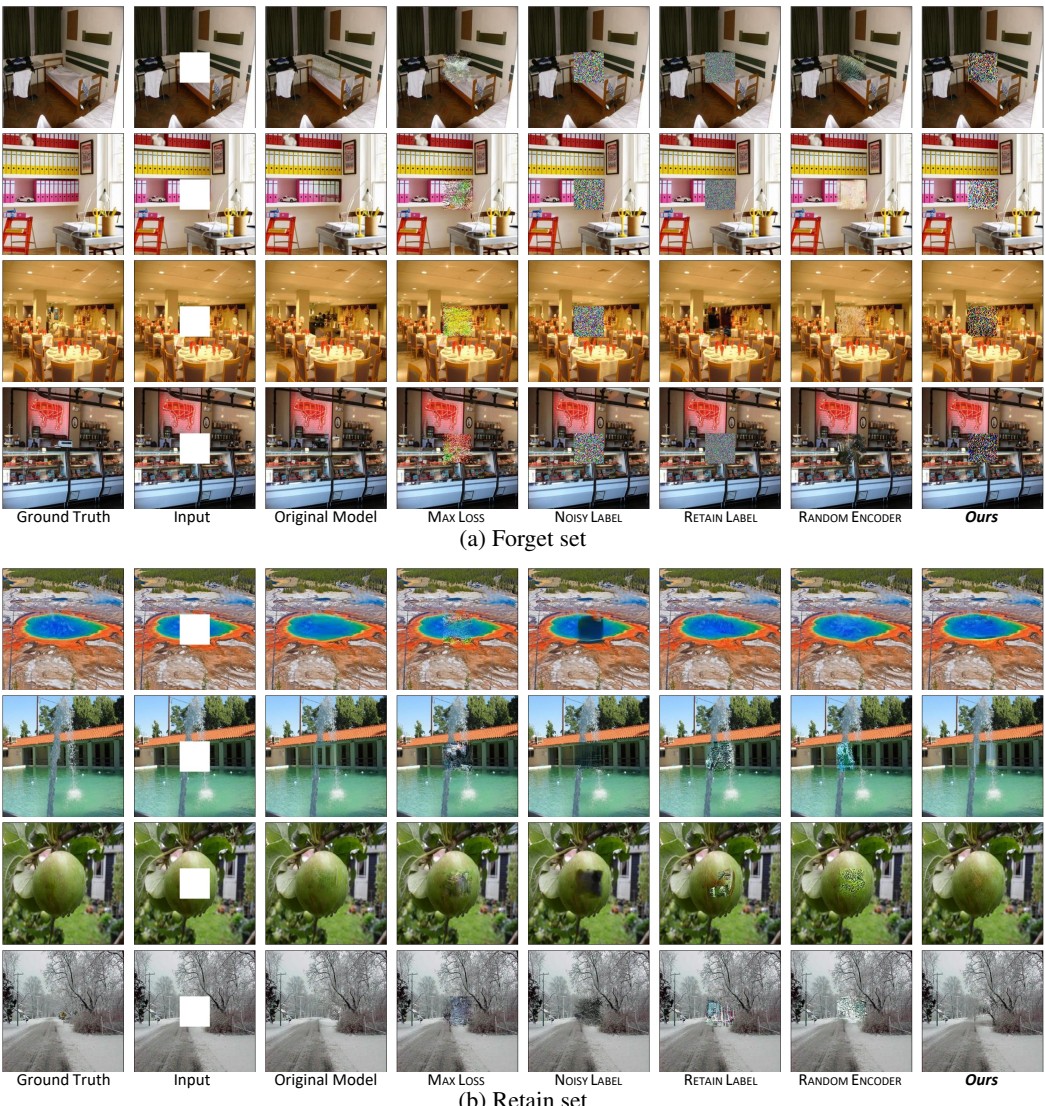

(a) Forget set

(b) Retain set

Figure D.7: Diffusion models: cropping $4 \times 4$ patches at the center of the image, where each patch is $16 \times 16$ pixels. As shown, our approach has almost identical performance as the original model (i.e., before unlearning); the generated images on forget set are some random noise. Overall, we outperform all other baseline methods.

## D.2 DIFFUSION

We provide the visualization results under center crop size of $4 \times 4$ patches in Fig. D.7. We also provide more visualization results under center crop size of $8 \times 8$ patches in Fig. D.8. As shown in Fig. D.7 and Fig. D.8, out approach can generate very high quality images on retain set while filling some random noise on the forget set. As a contrast, the other baselines methods struggle on the performance drop on the retain set. RANDOM ENCODER cannot even forget the information on the forget set.

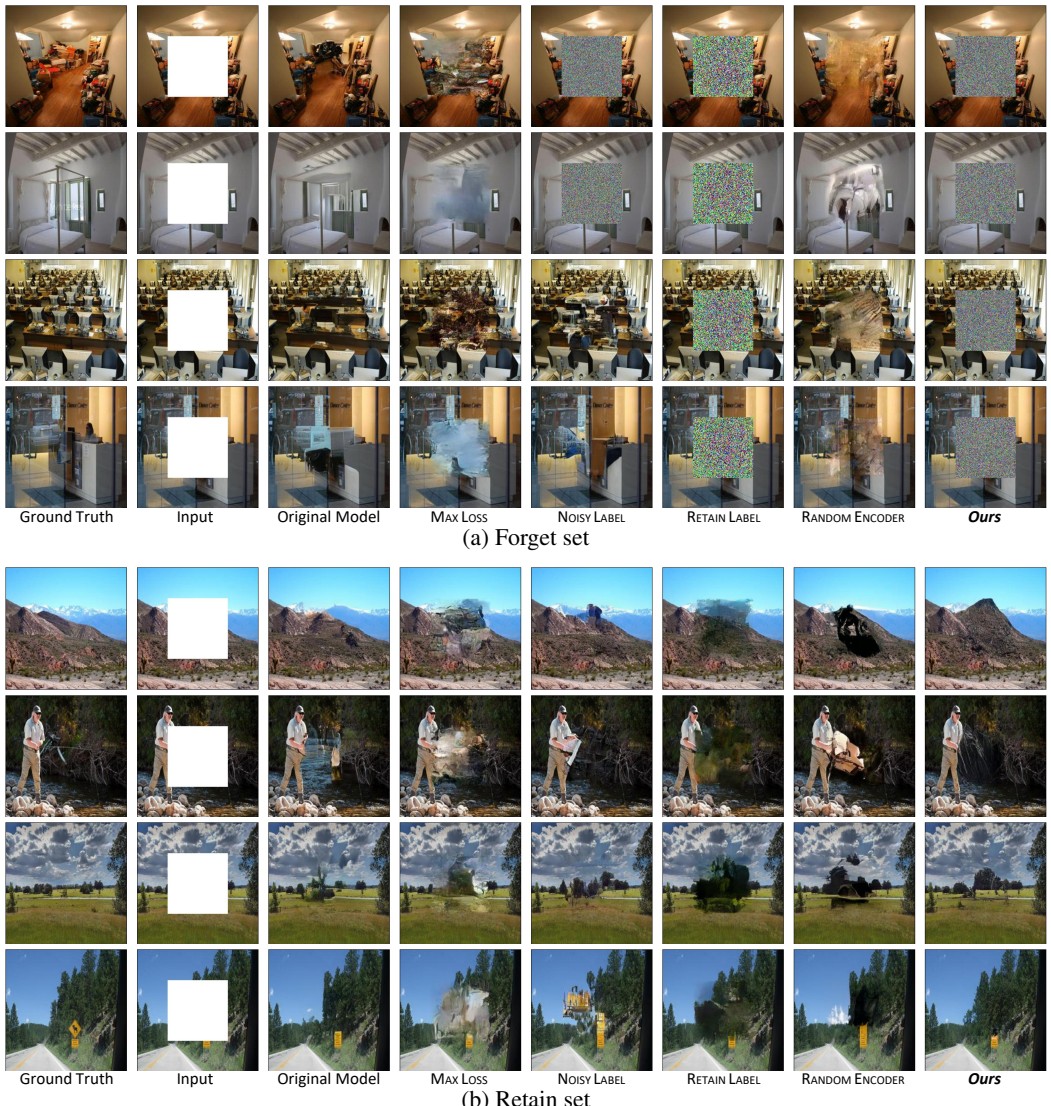

Figure D.8: Diffusion models: cropping $8 \times 8$ patches at the center of the image, where each patch is $16 \times 16$ pixels. As shown, our approach has almost identical performance as the original model (i.e., before unlearning); the generated images on forget set are some random noise. Overall, we outperform all other baseline methods.

## D.3 VQ-GAN

We provide the visualization results under center crop size of $4 \times 4$ patches in Fig. D.9. We also provide more visualization results under center crop size of $8 \times 8$ patches in Fig. D.10. As shown in Fig. D.9 and Fig. D.10, out approach can generate very high quality images on retain set while fill some random noise on the forget set. RETAIN LABEL cannot even forget the information on the forget set. Moreover, as shown in Fig. D.9, there are two type of fishes; one is in forget set and another one is in retain set (the fourth and fifth rows). Our approach can handle this subtle difference among difference classes.

## D.4 MAE

**Random Masking** In the original paper (He et al., 2022), MAE primarily focuses on the reconstruction of randomly masked images. Hence, besides the center cropping reported in the main paper, we also report the results under random masking. As shown in Fig. D.11 and Fig. D.12, the performance on the retain set of the

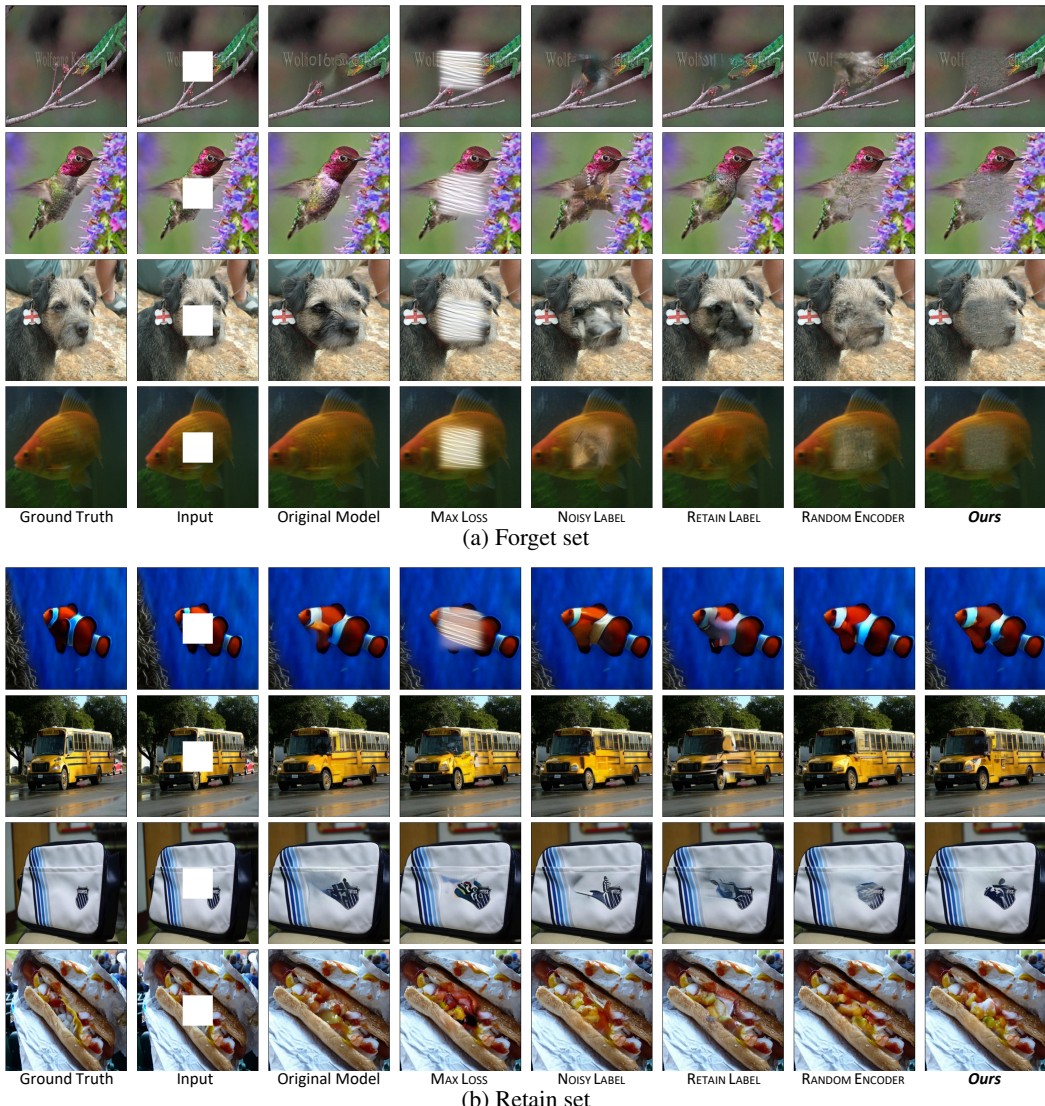

Figure D.9: VQ-GAN: cropping $4 \times 4$ patches at the center of the image, where each patch is $16 \times 16$ pixels. As shown, our approach has almost identical performance as the original model (i.e., before unlearning); the generated images on forget set are some random noise.

unlearned model is almost identical as original model. In contrast, there is a significant performance drop on the forget set.

# E    ABLATION STUDY

## E.1    VARYING CROPPING PATTERN AND RATIO

In the main paper, we primarily show the results of uncropping/inpainting tasks. We vary the cropping patterns and use the VQ-GAN to generate the images under these different patterns. We shown the generated images with downward extension in Fig. E.13, upward extension in Fig. E.14, leftward extension in Fig. E.15, rightward extension in Fig. E.16, and outpaint in Fig. E.17, respectively. As shown in these images, our method is robust to different cropping types.

As shown in Fig. E.18 and Fig. E.19, we also vary the input cropping ratio and report the FID and CLIP embedding distance. Clearly, our approach is robust to different cropping ratios.

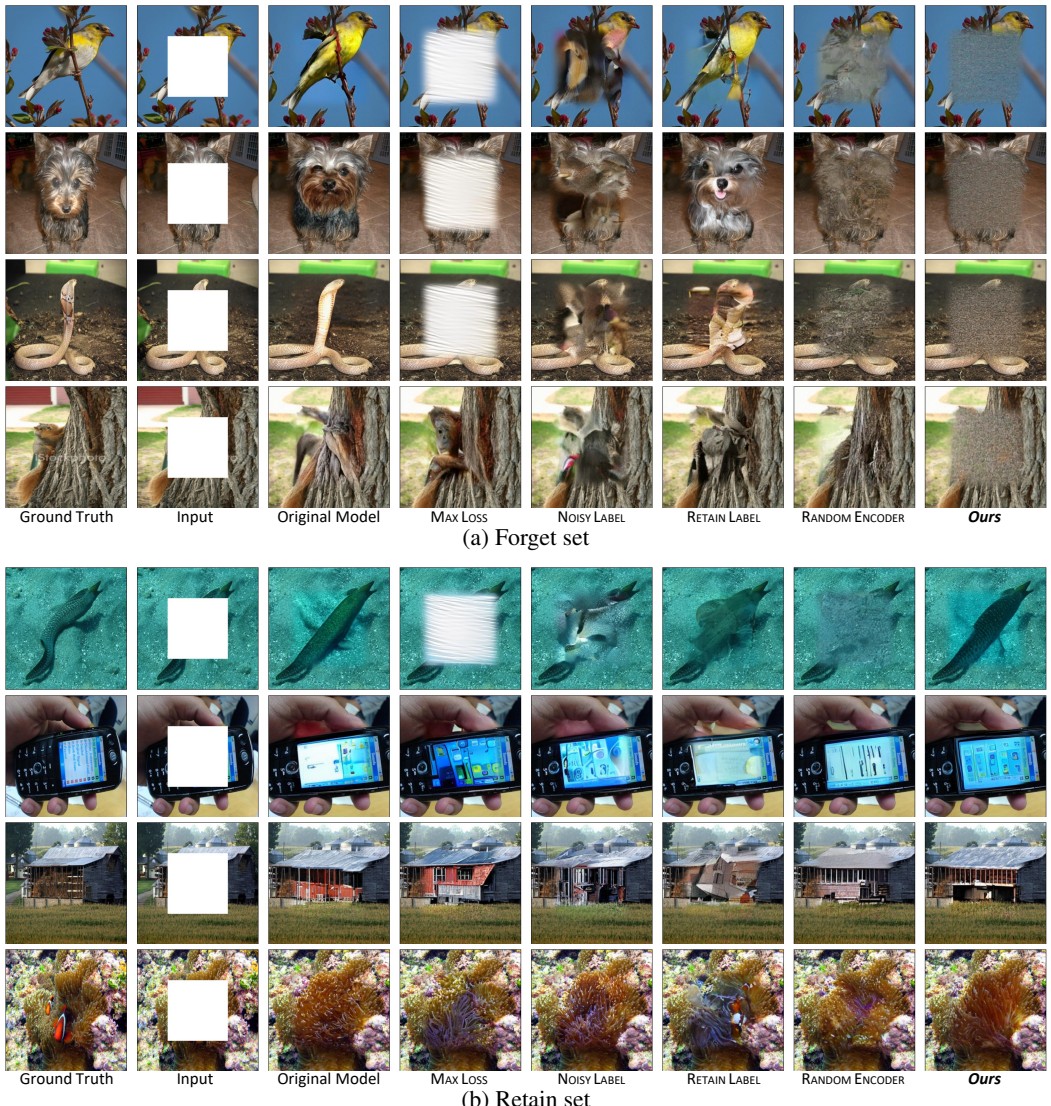

Figure D.10: VQ-GAN: cropping $8 \times 8$ patches at the center of the image, where each patch is $16 \times 16$ pixels. As shown, our approach has almost identical performance as the original model (i.e., before unlearning); the generated images on forget set are some random noise.

**Noise type.** We compare the performance of Gaussian Noise used in our method with uniform noise for VQ-GAN and MAE. Specifically, we set the noise in Eq. (10) as $n \sim \mathcal{U}[-1, 1]$ then conduct the unlearning under the same setup. We use the obtained models to generate 500 images for both retain set and forget set. We then compute multiple metrics for different noise types. As shown in Fig. E.20, Gaussian noise achieves better forgetting in the forget set in general (i.e., higher FID, lower IS and lower CLIP on forget sets). These results empirically verify the benefits of using Gaussian Noise, which coincides with our analysis (cf. Lemma 1).

## E.2 CROSS VALIDATION OF GENERATED IMAGES

To further verify the generated images do not have the information from forget set, we conducted the following experiment for VQ-GAN:

- We first conduct the unlearning for VQ-GAN for ImageNet to obtain the *unlearned model*, under the same setup mentioned in Section 4.

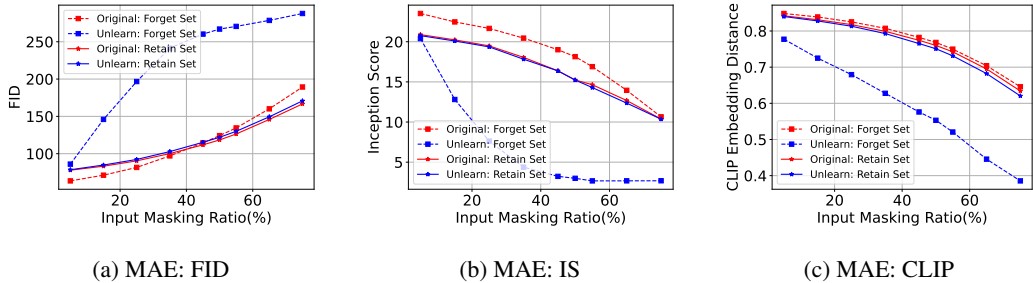

(a) MAE: FID  (b) MAE: IS  (c) MAE: CLIP

Figure D.11: The quality of reconstructed images given random masked images by MAE under varying random masking ratio; e.g., masking 128 out of 256 patches means 50% cropping ratio. We compare the original model and the unlearned model by our approach. As shown, the performance of these two models on the retain set are almost identical, while the unlearned model has a significant performance drop on the forget set. See Fig. D.4 for some examples of generated images.

Table E.4: The comparison of cross validation in Appendix E.2. We first use different models to do the inpainting tasks, i.e., reconstructing the central cropped patches (FIRST reconstructions). Given the FIRST reconstructions as input, we then use the original model (before unlearning) to do the outpainting tasks and get the SECOND reconstructions, i.e., re-generated the outer patches based on these reconstructed central patches from FIRST reconstructions. We then evaluate the quality of these SECOND reconstructed images under various metrics. Lower FID, higher IS, and higher CLIP values indicate higher image quality.

|  | FID | | IS | | CLIP | |
|---|---|---|---|---|---|---|
|  | $D_R$ | $D_F$ | $D_R$ | $D_F$ | $D_R$ | $D_F$ |
| Original Model | 36.14 | 28.96 | 27.28 | 27.95 | 0.37 | 0.49 |
| Unlearned Model | 36.74 | 156.68 | 26.62 | 7.67 | 0.37 | 0.28 |

- Second, given the center-cropped input images (cropping central $8 \times 8$ patches), we then use this *unlearned model* to generate the images on both forget set and retain set (i.e., image uncropping/inpainting). Here we call them FIRST generated images.

- Given these FIRST generated images, we only keep the reconstructed central $8 \times 8$ patches (cropping the outer ones) as the new input to the original VQ-GAN (i.e., the model before unlearning) and get the newly generated images (here we call them SECOND generated images). We then evaluate the quality of SECOND generated images for both forget set and retain set and report the results.

- We also conduct the above process for the *original model* (i.e., before unlearning) as the baseline/reference. The results are given below:

As shown in Table E.4 and Figure E.21, compared to the original model, given the *noise* (i.e., FIRST generated images on the forget set) from the unlearned model as the input, the SECOND generated images has very low quality in terms of all these three metrics. This means that the *noise* indeed are not correlated with the real forget images.

In contrast, the performance on the retain set is almost the same before and after unlearning. This indicates that our approach indeed preserves the knowledge on the retain set well.

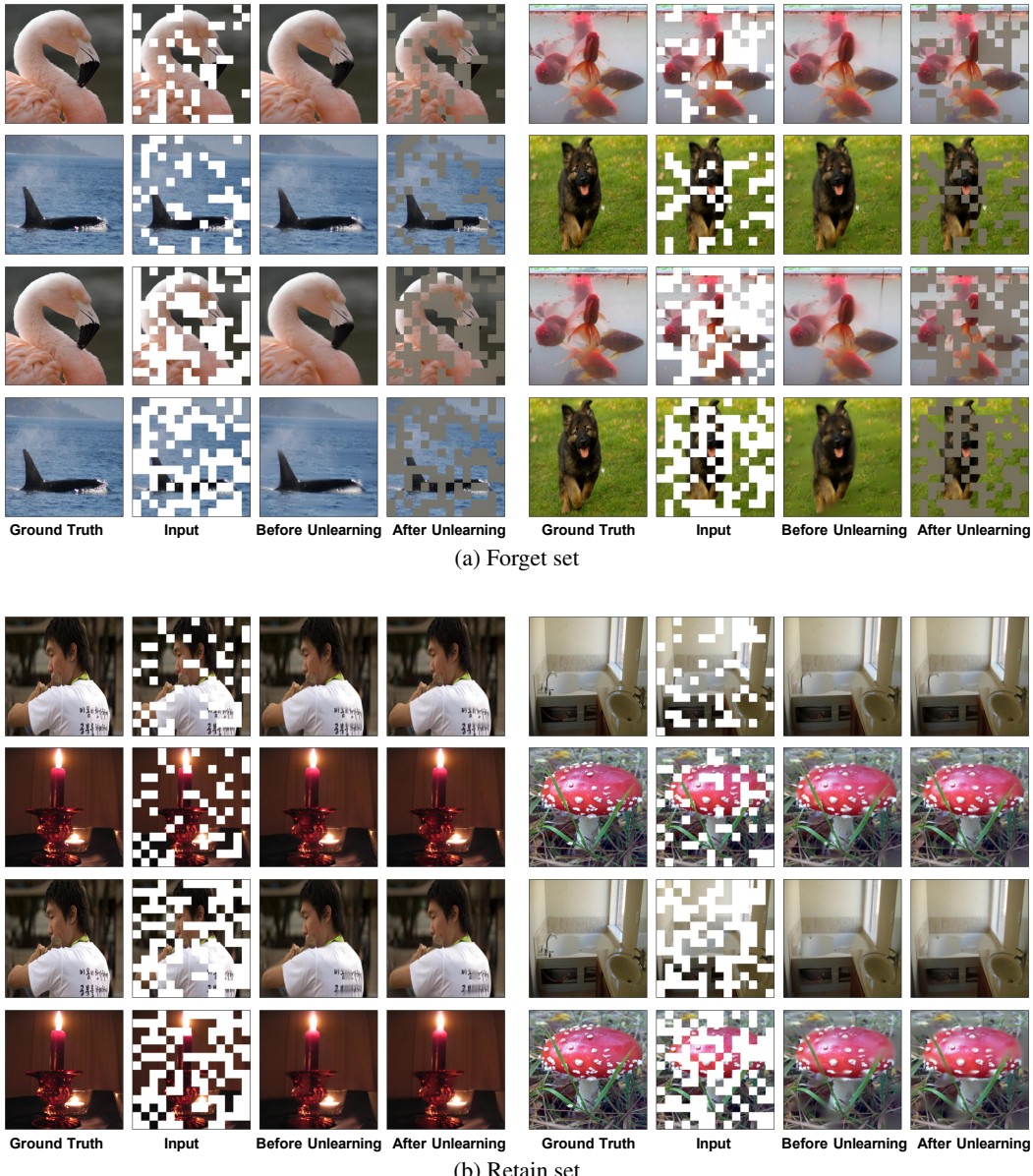

(a) Forget set

(b) Retain set

Figure D.12: Reconstruction of random masked images by MAE. For a single image, we test 25% masking ratio and 50% masking ratio.

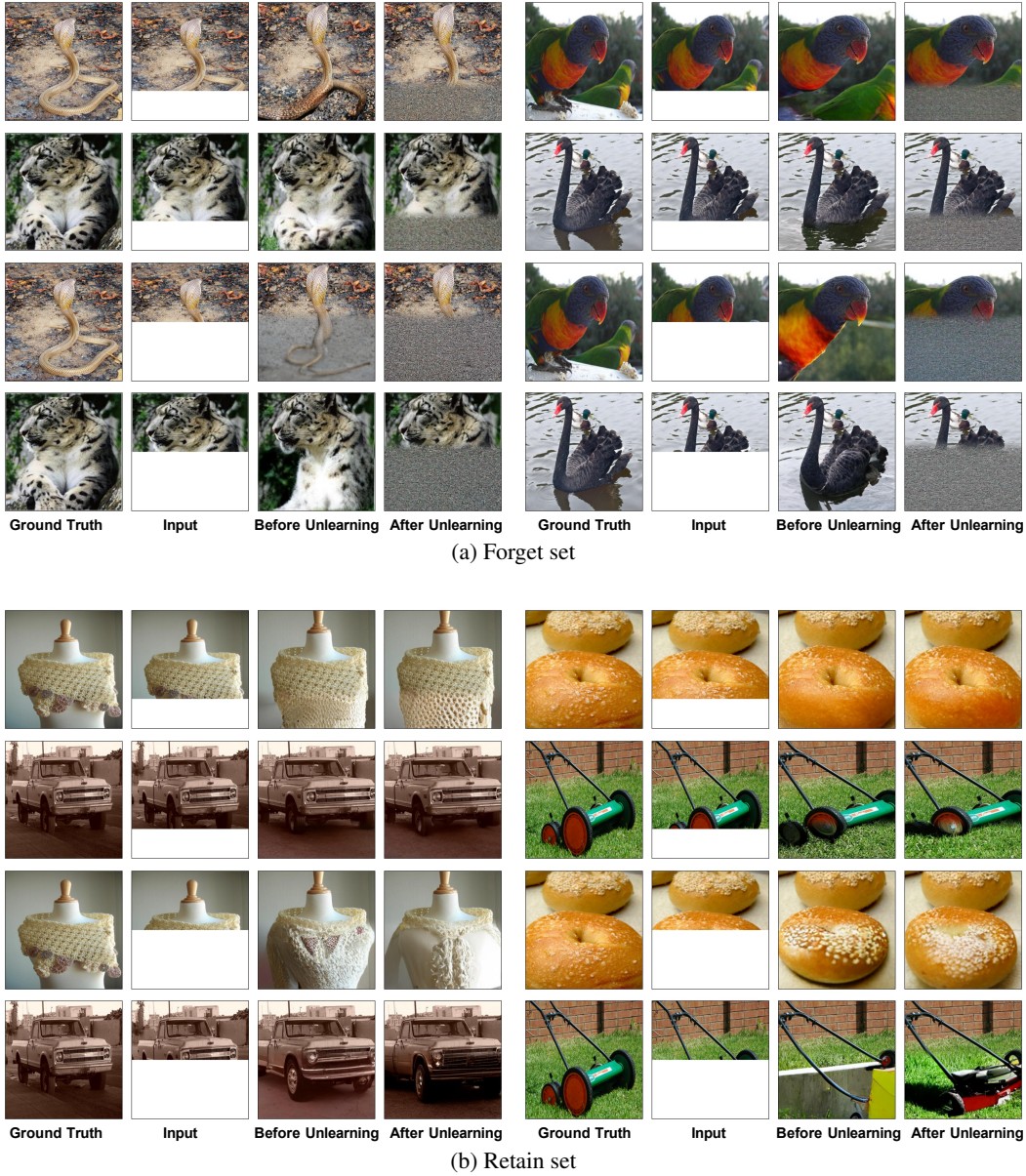

(a) Forget set

(b) Retain set

Figure E.13: Ablation study: Downward extension by VQ-GAN. We visualize he performance of the original model (before unlearning) and the obtained model by our approach (after unlearning). We show results on both forget set and retain set.

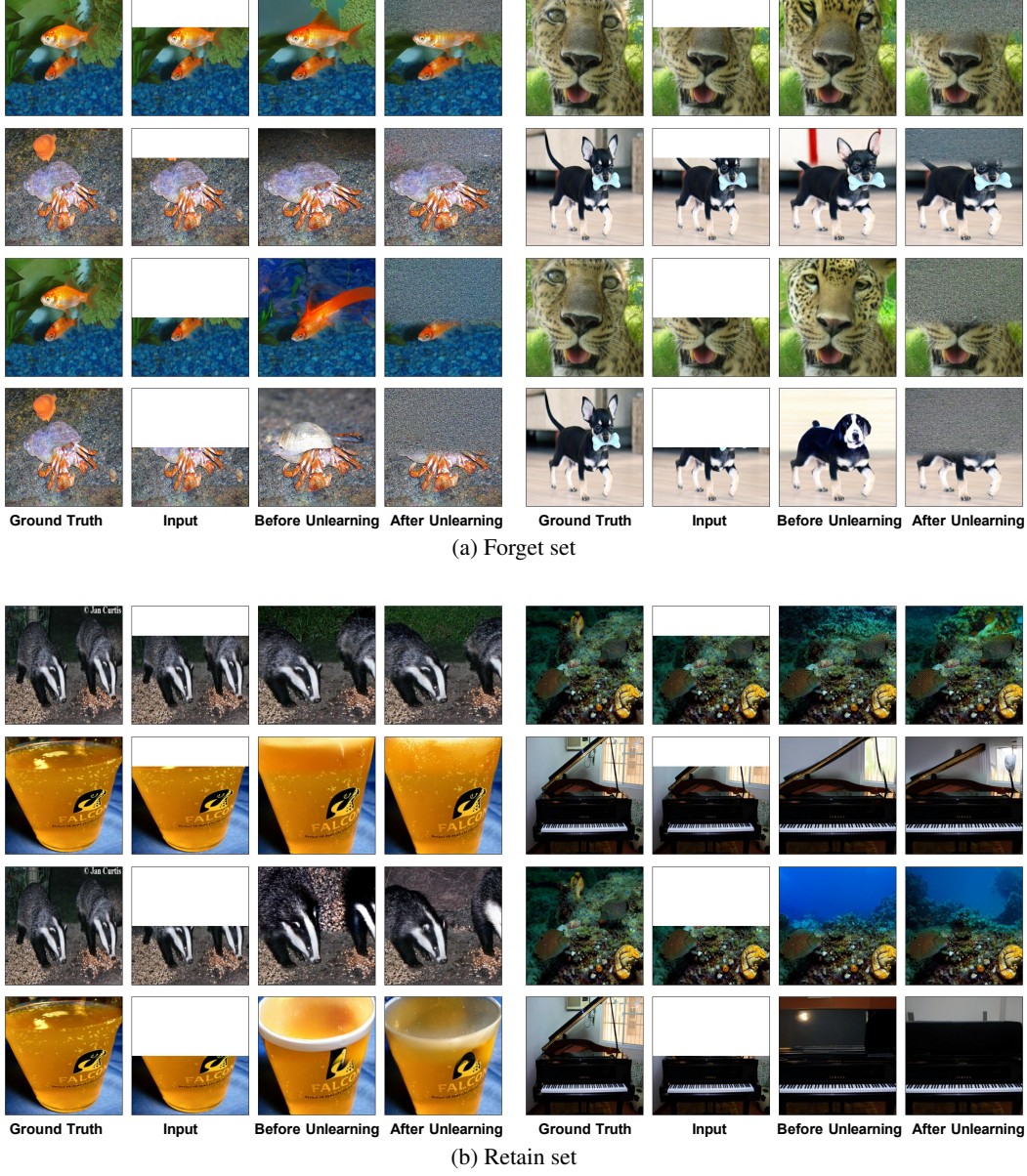

Figure E.14: Ablation study: Upward extension by VQ-GAN. We visualize he performance of the original model (before unlearning) and the obtained model by our approach (after unlearning). We show results on both forget set and retain set.

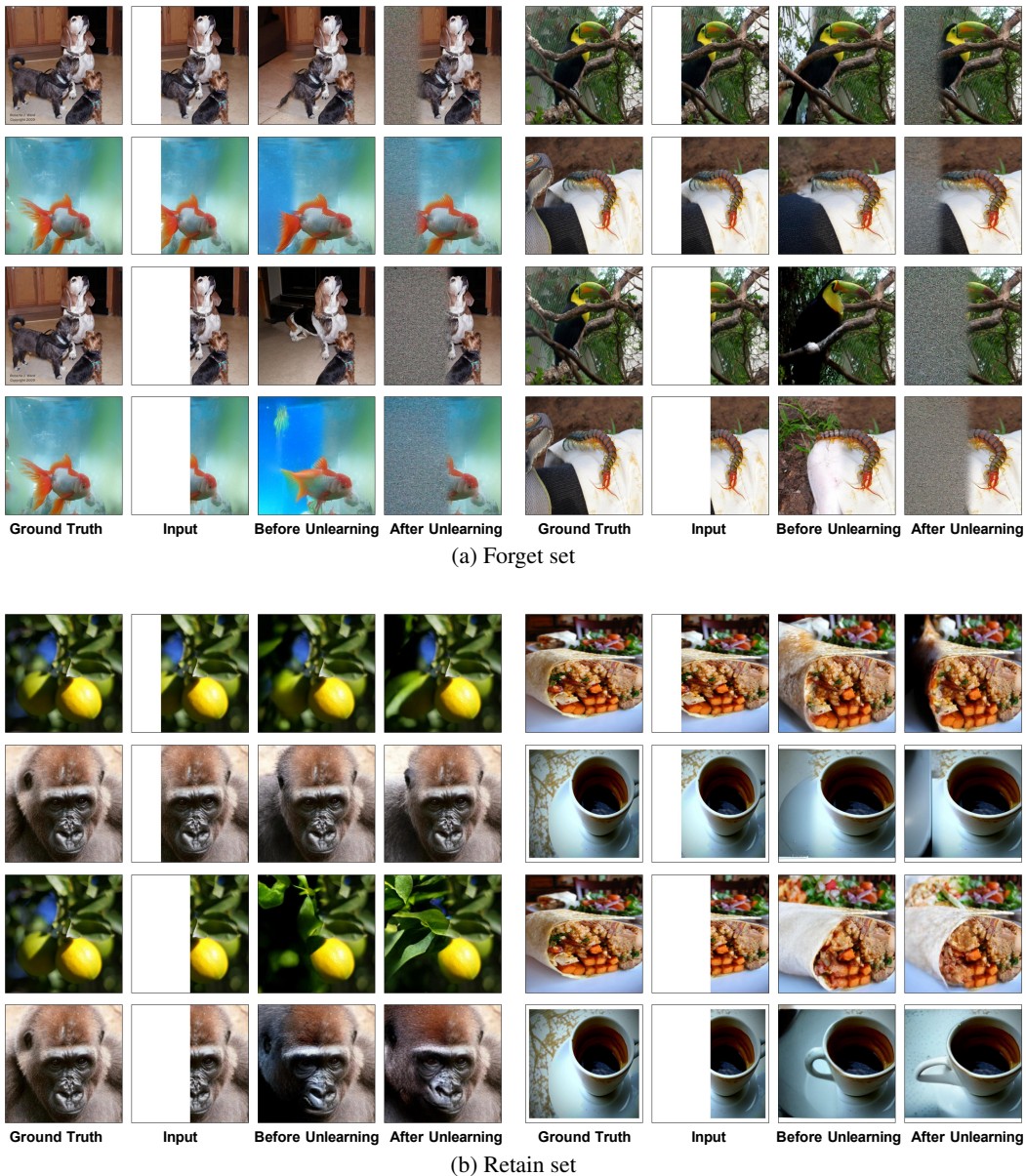

Figure E.15: Ablation study: Leftward extension by VQ-GAN. We visualize he performance of the original model (before unlearning) and the obtained model by our approach (after unlearning). We show results on both forget set and retain set.

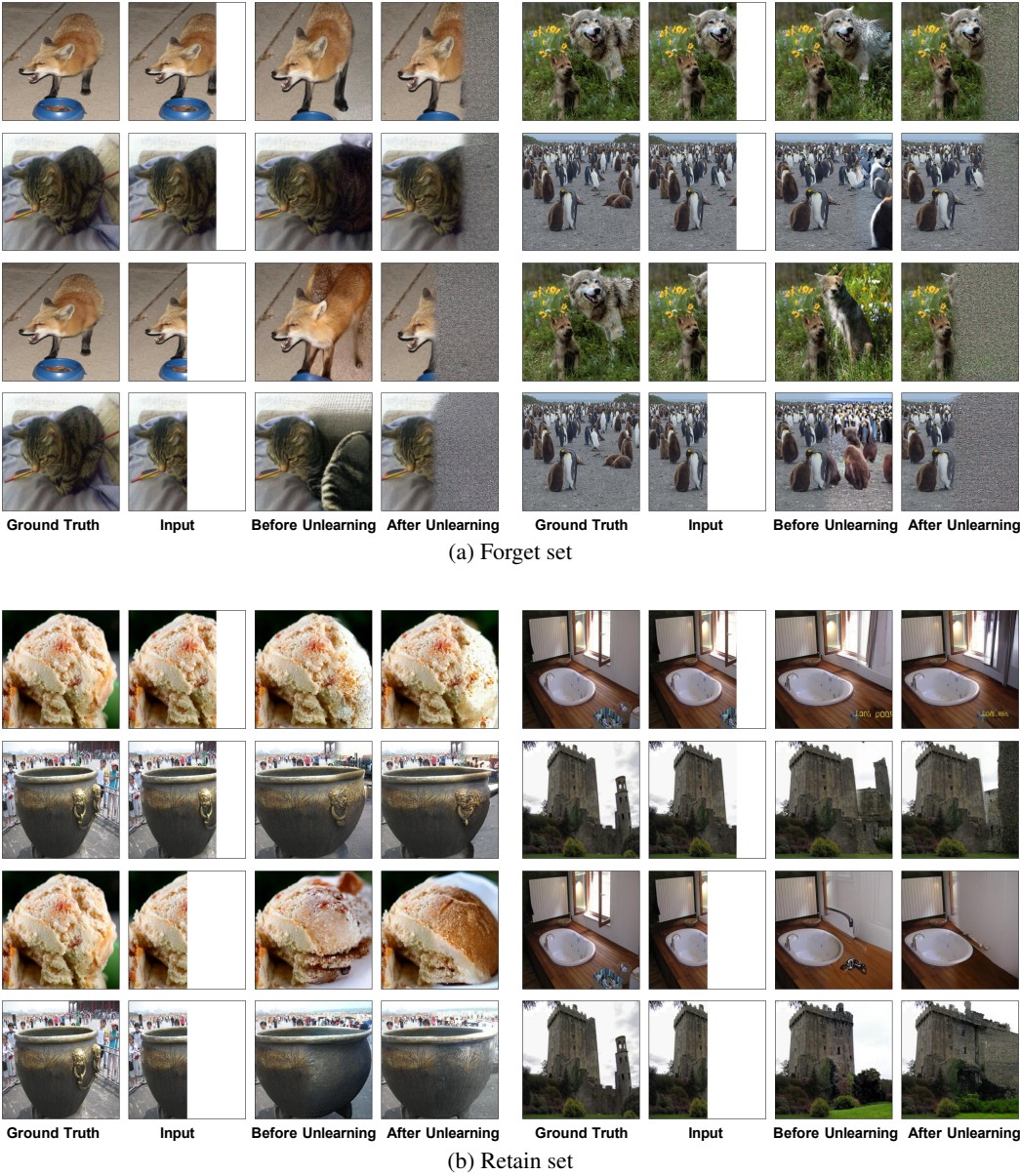

Figure E.16: Ablation study: Rightward extension by VQ-GAN. We visualize he performance of the original model (before unlearning) and the obtained model by our approach (after unlearning). We show results on both forget set and retain set.

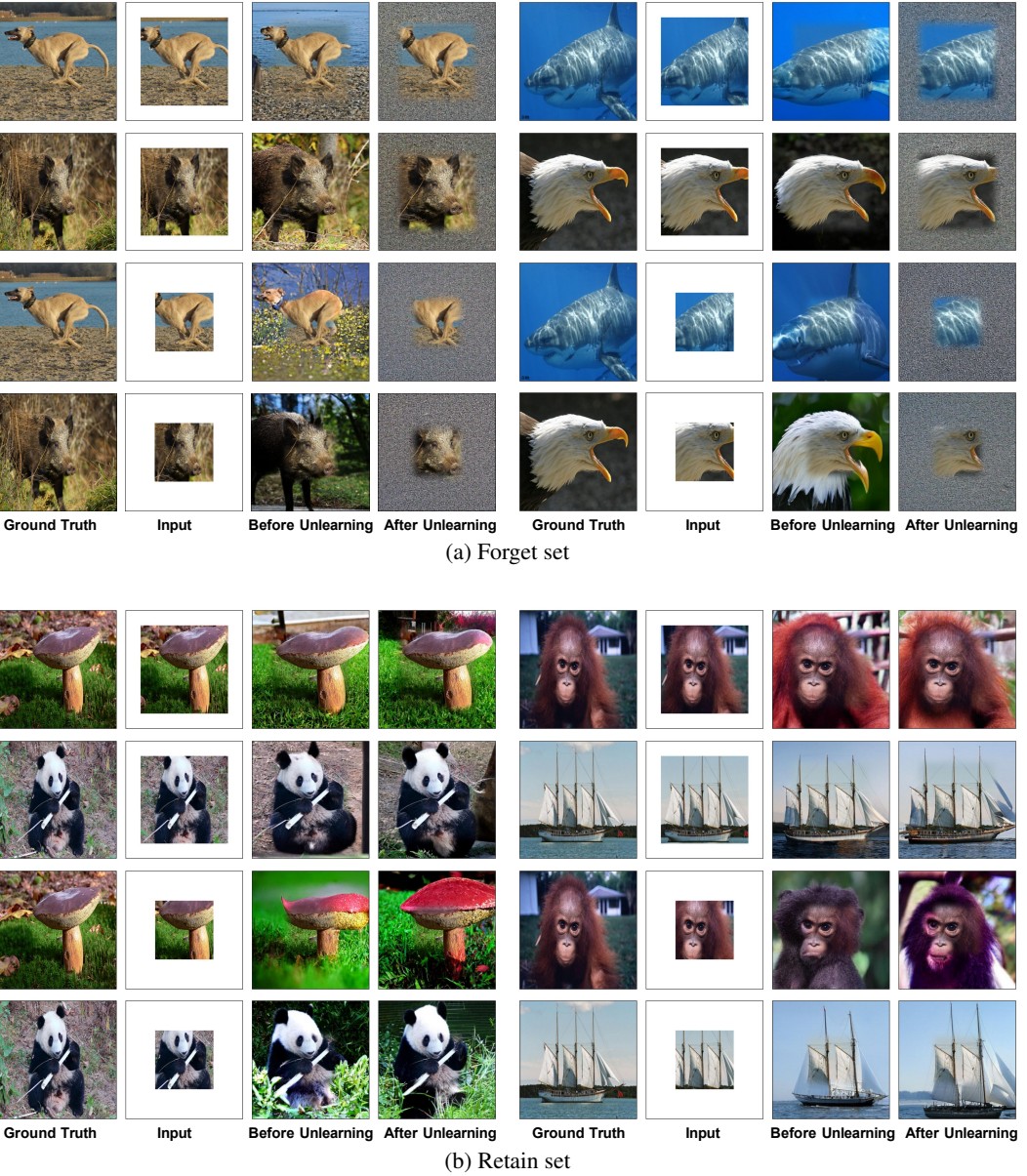

Figure E.17: Ablation study: Outpaint by VQ-GAN. We visualize he performance of the original model (before unlearning) and the obtained model by our approach (after unlearning). We show results on both forget set and retain set.

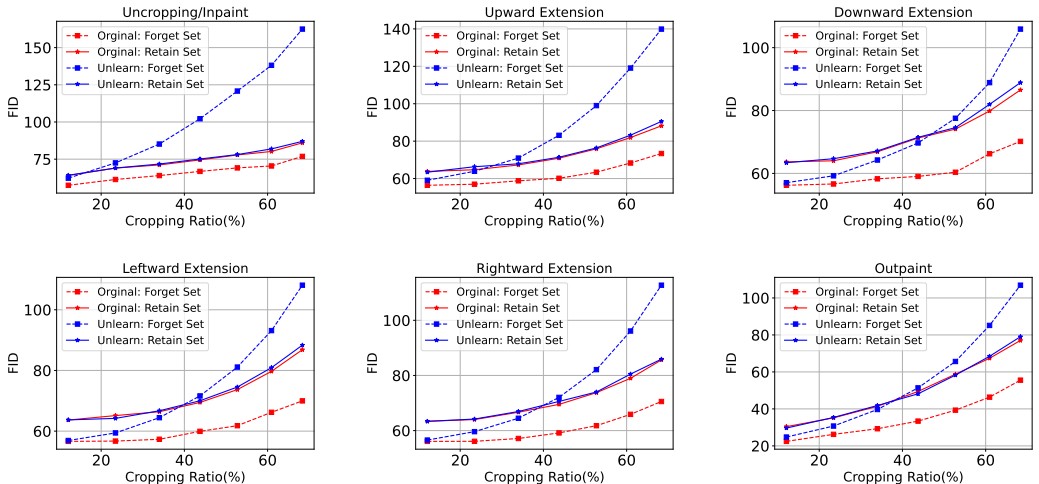

Figure E.18: Ablation study: FID *vs.* input cropping ratios under different I2I generation tasks.

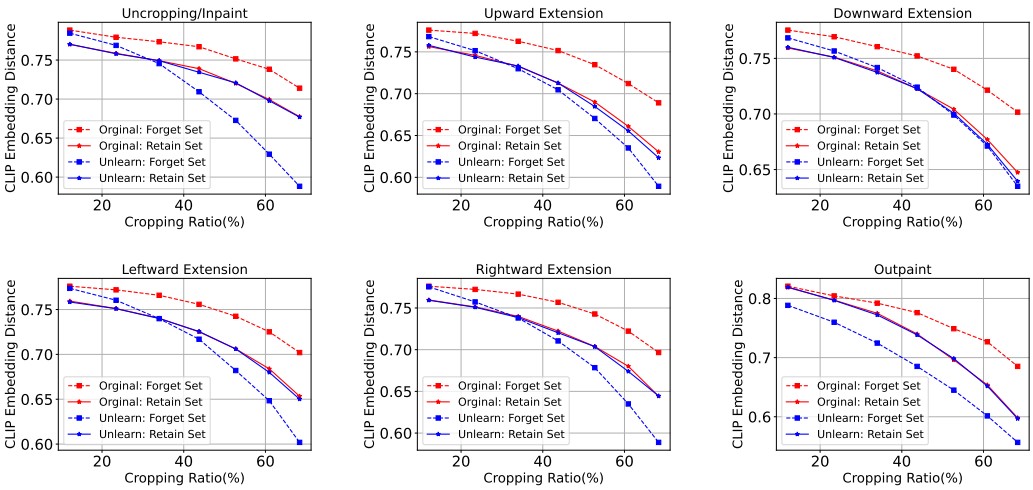

Figure E.19: Ablation study: CLIP embedding distance *vs.* input cropping ratios under different I2I generation tasks.

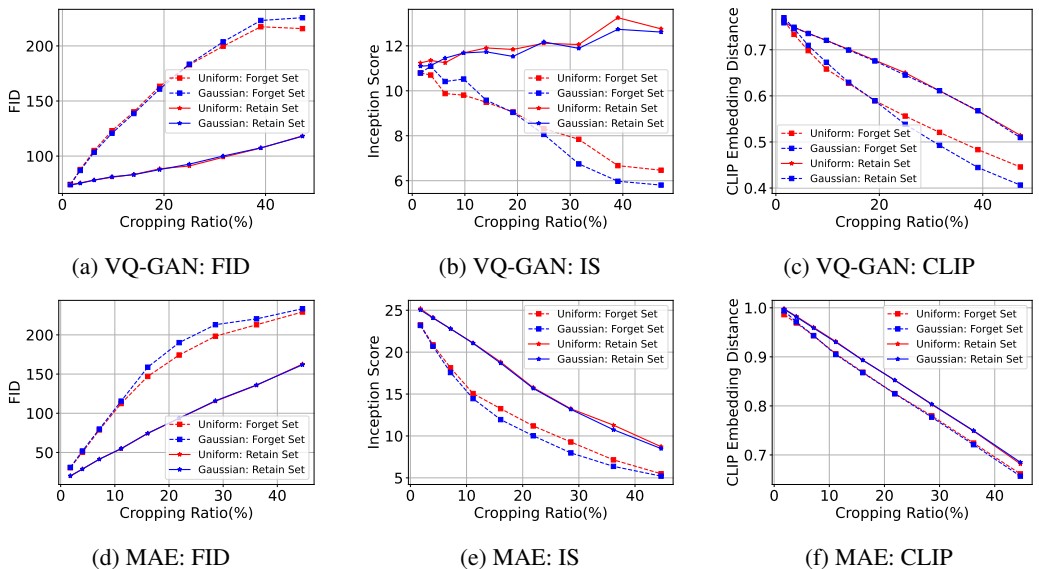

(a) VQ-GAN: FID      (b) VQ-GAN: IS      (c) VQ-GAN: CLIP

(d) MAE: FID      (e) MAE: IS      (f) MAE: CLIP

Figure E.20: Ablation study of noise type. We test the performance under varying ratios of central cropping; e.g., cropping $8 \times 8$ out of 256 patches means a 25% cropping ratio. For VQ-GAN and MAE, Gaussian noise achieves better forgetting in the forget set in general (i.e., it achieves higher FID, lower IS and lower CLIP on forget set).

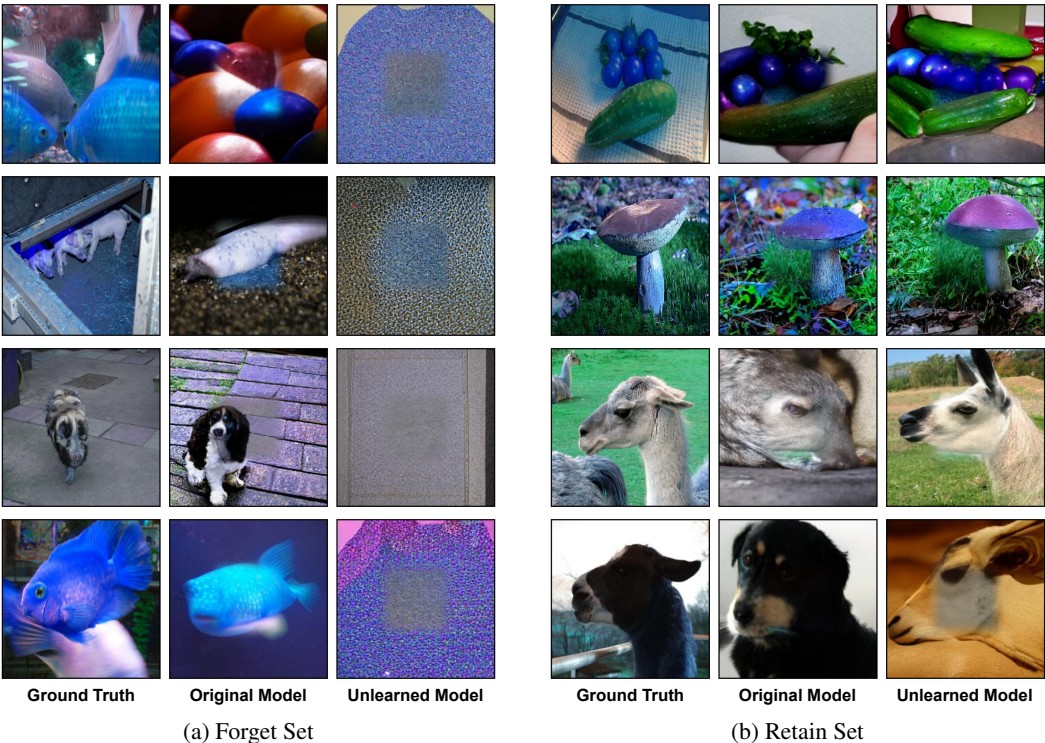

(a) Forget Set      (b) Retain Set

Figure E.21: Visualization of cross validation in Appendix E.2. We first use different models to do the inpainting tasks, i.e., reconstructing the central cropped patches (FIRST reconstructions). Given the FIRST reconstructions as input, we then use the original model (before unlearning) to do the outpainting tasks and get the SECOND reconstructions, i.e., re-generated the outer patches based on these reconstructed central patches from FIRST reconstructions. We then evaluate the quality of these SECOND reconstructed images under various metrics.

