# OpenReview forum: "Machine Unlearning for Image-to-Image Generative Models"
_ICLR.cc/2024/Conference — ICLR 2024 poster_

### Official Review · Reviewer_hK5e · 2023-10-19

**Soundness:** 3 good
**Presentation:** 3 good
**Contribution:** 3 good
**Rating:** 8
**Confidence:** 4

**Summary:**

This paper explores the concept of "machine unlearning," a method to deliberately forget data from models to comply with regulations. While existing methods have focused on classification models, this paper introduces a framework for machine unlearning in image-to-image generative models. It presents a computationally efficient algorithm supported by theoretical analysis that effectively removes information from "forget" samples without significant performance degradation on "retain" samples. The algorithm's effectiveness is demonstrated on large-scale datasets (ImageNet-1K and Places-365) and is notable for not requiring retain samples, aligning with data retention policies. This work is a pioneering effort in the systematic exploration of machine unlearning tailored for image-to-image generative models.

**Strengths:**

- The idea of using the E_0 to generate L_F and L_R is interesting and a smart way to overcome the problems of estimating MI.
- The theoretical analysis of the paper is well written and looks (beside of some remarks) sound to me.
- Having done so much experiments using different image datasets

**Weaknesses:**

I understand that the work was only done for I2I generation, which is a complex part already, however I was wondering if there is really no related work in this domain. I am not an expert in this domain but would propose to have a more in depth look at the literature for more related work.

**Questions:**

Abstract: „primarily focused on classification models, leaving the landscape of unlearning for generative models relatively unexplored.“: Is this true? Are there any references which support this statement?

Intro: „Informally speaking, we define a generative model as having “truly unlearned” an image when it is unable to faithfully reconstruct the original image when provided with only partial information (see Figure 1 for an illustrative example where the partial information involves center cropping2).“: I wonder if you did any analysis if the „noise“ is fully uncorrelated with the part of the image before the unlearning? Maybe there are still some high level corrections left. In learning fair representations people do train secondary models on the representations and try to predict the sensitivity attribute from the fair representations. I wonder if such study can be used to check if a secondary autoencoders could not re-generate the cropped image from the generated noise.

Equation 3: why is alpha not between 0-1 so you have (1-alpha) x term0 - alpha term1?

Theorem 1: Why is having the same decoder not a problem? Couldn’t the decoder not fully remember the images in some way or could you not extract some of the images by some information extraction techniques?

**Details Of Ethics Concerns:**

The work is directly related to privacy and legal compliance. The correctness of the method and the way how it gets used needs to be verified in detail for any potential application.

---

> ### Author Response · Authors · 2023-11-18
> **Response to Reviewer hK5e(1/2)**
>
> We really appreciate your encouragement for our work. Please see our responses to your questions below.
>
>
> **Question 1** Abstract: “primarily focused on classification models, leaving the landscape of unlearning for generative models relatively unexplored.“: Is this true? Are there any references which support this statement?
>
> **Response 1** We provide the following categories of references to support our claim that *’the landscape of unlearning for generative models is relatively unexplored’*:
>
> - Many recent review/survey papers on machine unlearning. As shown in [1,2,3,4], there are only three papers targeting unlearning for generative models out of hundreds of cited papers in these papers.
>
> - Open-source collections of machine unlearning papers [5]. Out of 254 papers shown in this collection, only eight papers focus on the unlearning for language generation and only 9 papers target the unlearning for image generation. Specifically, for unlearning on image generation models, most papers only explore GANs and none of them addresses the image-to-image generation. In contrast, our approach is generally applicable to GANs, MAE and diffusion models and is the first to unlearn on image-to-image generative models.
>
> **Question 2** Intro: "Informally speaking, we define a generative model as having 'truly unlearned' an image when it is unable to faithfully reconstruct the original image when provided with only partial information (see Figure 1 for an illustrative example where the partial information involves center cropping2).": I wonder if you did any analysis if the "noise" is fully uncorrelated with the part of the image before the unlearning? Maybe there are still some high-level corrections left. In learning fair representations people do train secondary models on the representations and try to predict the sensitivity attribute from the fair representations. I wonder if such study can be used to check if a secondary autoencoders could not re-generate the cropped image from the generated noise.
>
> **Response 2** We appreciate for this valuable suggestion for a more thorough evaluation of our approach. To this answer this question, we conducted the following experiment for VQ-GAN:
>
> - We first conduct the unlearning for VQ-GAN for ImageNet to obtain the *unlearned model*, under the same setup mentioned in Section 4.1.
>
> - Given the center-cropped input images (cropping central $8\times 8$ patches), we then use this *unlearned model* to generate the images on both forget set and retain set (i.e., image upcropping/inpainting). Here we call them *FIRST* generated images.
> - Given these *FIRST* generated images, we only keep the reconstructed central $8\times 8$ patches (cropping the outer ones) as the new input to the original VQ-GAN (i.e., the model before unlearning) and get the newly generated images (here we call them *SECOND* generated images). We then evaluate the quality of *SECOND* generated images for both forget set and retain set and report the results.
> - We also conduct the above process for the *original model* (i.e., before unlearning) as the baseline/reference. The results are given below:
>
> | |FID: $D_R$|FID: $D_F$|IS: $D_R$|IS: $D_F$|CLIP: $D_R$|CLIP: $D_F$|
> |:-:|:-:|:-:|:-:|:-:|:-:|:-:|
> |Original Model|36.14|28.96|27.28|27.95|0.37|0.49|
> |Unlearned Model|36.74|156.68|26.62|7.67|0.37|0.28|
>
> > **NOTE:**  Lower FID, higher IS, and higher CLIP values indicate higher image quality.
>
> As shown above, compared to the original model, given the *noise* (i.e., *FIRST* generated images on the forget set) from the unlearned model as the input, the *SECOND* generated images has very low quality in terms of all these three metrics. This means that the *noise* indeed are not correlated with the real forget images.
>
> In contrast, the performance on the retain set is almost the same before and after unlearning. This indicates that our approach indeed preserves the knowledge on the retain set well.
>
> We will add these results and provide some visualizations in the revised paper.

---

> > ### Author Response · Authors · 2023-11-18
> > **Response to Reviewer hK5e(2/2)**
> >
> > **Response 3** We would like to point out that these two methods are mathematically equivalent, in principle: $L_R+\alpha L_F$ and $(1-\alpha) L_R+\alpha L_F$. Specifically, we can rewrite $L_R+\alpha L_F$ as:
> >
> > $L_R+\alpha L_F=(1+\alpha) (\frac{1}{1+\alpha}L_R+ \frac{\alpha}{1+\alpha} L_F) = (1+\alpha) [(1-\beta) L_R+ \beta L_F]$
> >
> > where $\beta=\frac{\alpha}{1+\alpha}$. Therefore, the main difference between $(1+\alpha) [(1-\beta) L_R+ \beta L_F]$ and $(1-\alpha) L_R+\alpha L_F$ is that there’s a scaling factor $(1+\alpha) $ for the former one. In practice, one can use either model based on personal preference.
> >
> > **Question 4** Theorem 1: Why is having the same decoder not a problem? Couldn’t the decoder not fully remember the images in some way or could you not extract some of the images by some information extraction techniques?
> >
> > **Response 4** This is a very insightful question. In fact, this kind of information extraction technique is highly correlated to another  important problem called Model Inversion Attacks (MIA) [6,7]. Briefly speaking, MIA tries to generate samples that are very similar to the original training samples.
> >
> > Currently, our approach cannot guarantee the robustness to MIA; in fact, even unlearning the entire model may still suffer from the vulnerability of MIA. We remark that like the unlearning of generative models, MIA for generative models, especially for diffusion models, is also an emerging problem [6,7]. We plan to explore the defense methods of the MIA in our future work .
> >
> > Having said that, we also want to highlight a couple of important benefits that result from  focusing on the encoder only vs. unlearning the entire model:
> >
> > - Efficiency: training the encoder only is much more efficient (and requires much less computational capability) compared to training the entire model.
> >
> > - Universal applicability. Unlearning the encoder enables us to propose a universal MSE-based loss instead of designing the different loss for distinct types of generative models.
> >
> >
> >
> > Besides the responses to your questions, we are kind of confused about your comment related to **Flag For Ethics Review**given the principle described in ICLR website (https://iclr.cc/public/CodeOfEthics). We kindly request the Reviewer elaborate more on this matter.
> >
> > We hope our responses address your concerns. We look forward to further discussions if you have any other questions.
> >
> >
> >
> > [1] Xu, Jie, et al. "Machine Unlearning: Solutions and Challenges." arXiv preprint arXiv:2308.07061 (2023).
> >
> > [2] Shaik, Thanveer, et al. "Exploring the Landscape of Machine Unlearning: A Survey and Taxonomy." arXiv preprint arXiv:2305.06360 (2023).
> >
> > [3] Xu, Heng, et al. "Machine Unlearning: A Survey." arXiv preprint arXiv:2306.03558 (2023).
> >
> > [4] Nguyen, Thanh Tam, et al. "A survey of machine unlearning." arXiv preprint arXiv:2209.02299 (2022).
> >
> > [5] https://github.com/jjbrophy47/machine_unlearning
> >
> > [6] Liu, Rongke. "An Indefensible Attack: Label-Only Model Inversion via Conditional Diffusion Model." arXiv preprint arXiv:2307.08424 (2023).
> >
> > [7] Carlini, Nicolas, et al. "Extracting training data from diffusion models." 32nd USENIX Security Symposium (USENIX Security 23). 2023.

---

### Official Review · Reviewer_4Xrc · 2023-11-01

**Soundness:** 3 good
**Presentation:** 3 good
**Contribution:** 3 good
**Rating:** 6
**Confidence:** 4

**Summary:**

This proposes a computationally-efficient unlearning approach, that demonstrates negligible performance degradation on the retain samples, while effectively removing the information from the forget samples. The authors also provide rigorous theoretical analysis for this image-to-image generative model. Empirical studies on two large-scale datasets, ImageNet-1K and Places-365 show the advantages on image-to-image task. Overall, it is good paper. The main concern is how to get the forget set and how to set the hyper parameter $\sigma$?

**Strengths:**

Quality/Clarity: the paper is well written and easy to follow. And both the experimental results and theoretical analysis demonstrate its advantages.

Originality/significance: the algorithm is new and underpinned by rigorous theoretical analysis. It systematically explores machine unlearning for I2I generative models.

**Weaknesses:**

1. The main concern is that the paper assumption is ideal and its hyper parameter \sigma, which may not hold in practice.
2. From Lemma 1, $\sigma$ should be from the forget set? In the appendix C.4, however it sets $\sigma=I$, which is not estimated from forget set?

**Questions:**

The main concern is that the paper assumption is ideal and its hyper parameter \sigma, which may not hold in practice.

---

> ### Author Response · Authors · 2023-11-16
> **Response to Reviewer 4Xrc**
>
> We thank the reviewer for the feedback and encouragement! We address your question point-by-point below.
>
>
> **Question** The main concern is that the paper assumption is ideal and its hyper parameter $\Sigma$, which may not hold in practice.
>
>
>
> **Response** Regarding the assumptions used in our paper, we tried our best in making them align with practice. Specifically, one of the main assumptions is that the original model can do a perfect reconstruction for the cropped images. This assumption is somehow ideal, yet practically reasonable, especially given the recent progress in diffusion models [1,2]. We also note that since our work is the first to target unlearning for image-to-image generative models, we hope it will pave the way for follow-up works that will relax some of our assumptions.  The theoretical results we provide offer a solid basis for such future extensions.
>
> We also agree with Reviewer that, ideally, we should use the exact $\Sigma$ of the images from the forget set. However, there are some computational barriers to using the exact $\Sigma$ for a high-resolution image dataset. Specifically, consider a commonly used 256$\times$256 resolution for image generation tasks, the distribution of the generated images will have $256\times 256\times 3 \approx 2\times 10^5$ dimensions. The size of the covariance matrix $\Sigma$ for such a high-dimensional distribution is around $(2\times 10^5)^2=4\times 10^{10}$, which requires around 144GB memory if stored in float precision thus is not practical.
>
>
> Consequently, to address the computational barrier of $\Sigma$, we use some approximated methods derived from some empirical observations on some small datasets.
> Specifically, we compute the exact $\Sigma$ for some small-scale image dataset, including MNIST and CIFAR10/100.
>
> - *Off-diagonal elements*: as shown in Figure C.6 at Appendix C.4, the values on the diagonal of exact $\Sigma$ for these datasets are generally much larger than the off-diagonal elements; hence, we truncate the off-diagonal elements to zero.
> - *Diagonal elements*: since the input images for deep networks are typically normalized, we may set the diagonal elements to one.
>
> In short, using $I$ to approximate $\Sigma$ is a practical approximation alternative due to the extremely high computational costs of $\Sigma$ for high-resolution images.
> For future work, given our theoretical analysis, we believe that our approach will achieve better results (lower image quality) on forget set if we can find a way to use the exact $\Sigma$. Hence, we plan to explore the potential to reduce the computation of exact $\Sigma$ with low-rank approximation thus enabling the use of more accurate data-driven $\Sigma$.
>
> We will add the above clarifications to the revised paper. We hope our response can address your concerns. We look forward to further discussions if you have any other questions.
>
> [1] Song, Bowen, et al. "Solving inverse problems with latent diffusion models via hard data consistency." arXiv preprint arXiv:2307.08123 (2023).
>
> [2] Wallace, Bram, Akash Gokul, and Nikhil Naik. "Edict: Exact diffusion inversion via coupled transformations." Proceedings of the IEEE/CVF Conference on Computer Vision and Pattern Recognition. 2023.

---

> ### Author Response · Authors · 2023-11-22
> **Kindly expecting further discussions**
>
> Dear Reviewer 4Xrc,
>
> Thanks again for constructive feedback on our paper. We hope that our previous responses have successfully addressed your concerns. We would like to remind you that the author-reviewer discussion period is closing very soon. If there are any remaining questions or concerns, we are looking forward to further discussions!
>
> Best regards,
>
> The Authors of Paper 2859

---

### Official Review · Reviewer_L8mN · 2023-11-03

**Soundness:** 2 fair
**Presentation:** 2 fair
**Contribution:** 2 fair
**Rating:** 5
**Confidence:** 3

**Summary:**

This work proposes a mechanism for machine unlearning for image-to-image generative models. The work proposes a simple framework, minimizing the $l_2$ loss between embeddings from the forget training set and normal distribution, and maintaining the $l_2$ loss for the retain set embeddings. The results are provided on VQ-GAN, Diffusion Models and Masked Autoencoders on Imagenet-1K and Places-365 dataset comparing to reasonable baselines where the method shows good performance on retain set, while performance on forget set deteriorates as expected.

**Strengths:**

1. This work is focused on a timely and important problem as there is more discussing about regulating generative AI. For the current lawsuits facing several companies for their data practices on generative models and upcoming laws surrounding data retention policies, machine unlearning approaches may offer a possible solution.
2. The experimental across different models and datasets are thorough, and helpful for future work. The work shows comparisons across several reasonable baselines, across different image-to-image generative models. The results are shown across different relevant metrics such as FID, Inception Score and CLIP Distance.

**Weaknesses:**

1. It's unclear if the machine unlearning setup considered in this work is practical. For machine unlearning setups, the gold standard to mimic is a model trained only on the retain set. In this work, the approach performs this task by minimizing the embedding distance for the forget set to normal distribution. Does this lead to overall worse performance than the gold standard model (where only retain set is used to train from scratch)? It would be good to include this as a baseline for comparison.
2. The proposed approach itself is not too novel, while the work may have applied it first on image-to-image generative models.  This approach can be thought of a simple extension student-teacher or continual learning framework, and similar ideas have been explored in classification literature [1, 2].
3. It's unclear if the model shows strong performance throughout. While the performance on the retain set stays strong, it's unclear if **worst** performance on the forget set is a good metric. This again focuses on going back to the first point, the unlearning paradigm should only be focused on achieving performance of model trained *only* on retain set. Thus, if the unlearned model performs much worse on the forget set than the model trained only on the retain set this may not be a good metric.

[1] Zhang, Xulong, et al. "Machine Unlearning Methodology base on Stochastic Teacher Network." _arXiv preprint arXiv:2308.14322_ (2023).
[2] Zhang, Yongjing, et al. "Machine Unlearning by Reversing the Continual Learning." _Applied Sciences_ 13.16 (2023): 9341.

**Questions:**

1. The experimental setup for retain sample availability is not quite clear to me. Why are retain samples selected from the remaining 800 random classes? Also, this confused me if all the 1000 classes, or only 200 classes are used for the main experiments to train the original model.

---

> ### Author Response · Authors · 2023-11-16
> **Response to Reviewer L8mN (1/2)**
>
> We thanks for your constructive feedback for our work. We address the questions point-by-point below.
>
>
>
> **Question 1**: It's unclear if the machine unlearning setup considered in this work is practical. For machine unlearning setups, the gold standard to mimic is a model trained only on the retain set. In this work, the approach performs this task by minimizing the embedding distance for the forget set to normal distribution. Does this lead to overall worse performance than the gold standard model (where only retain set is used to train from scratch)? It would be good to include this as a baseline for comparison.
>
>
> **Response 1**: This is a very fundamental question, and we understand the reviewer’s concern to the definition/setup of our unlearning approach. Indeed, we did ask the same questions ourselves. First, we agree that most of previous works consider the training only on the retain set is the gold baseline for *classifications tasks*. However, recently it has been shown that this golden standard bear several concerns:
>
> - Some recent works show that for *‘some unlearning algorithms can generate models indistinguishable from the golden baseline beyond any arbitrarily small threshold while still exposing the deleted data’* for some cases [1]. In other words, the indistinguishability of training on retain set only cannot guarantee the data has been totally removed from the model, which is the goal in our paper.
>
>
> - Retraining these generative models from scratch on retain set is completely beyond our computation capacity. For the three models evaluated in our paper, it takes more than 10K GPU hours ($>$50 days on our 8-GPU server; estimated from the experimental setup in [2,3,4])  to train the model from scratch.
>
> - Given these considerations, our setup/definition primarily focuses on the practical scenarios other than the indistinguishability from the retrained model. For example, if the training set for the original model has some pornographic images by accident, our methods can make the unlearned model avoid generating pornographic images in a very efficient way other than expensive retraining the model from scratch. This way, the model's developers can avoid the potential violation of contents control. In short, our main goal is making the unlearned model not generate the samples from the forget set without hurting the performance on the retain set. Under this perspective, the performance drops on the forget set shown in our results is expected and dedicatedly designed. Moreover, the results show that on retain set, our approach has negligible performance drop compared to the model before unlearning.
>
> As for the performance comparison with the golden standard model, our method will have worse performance **on the forget set** than this golden baseline (the performance on the retain set is expected to be similar to the golden standard model), which is dedicatedly designed by our approach.
>
> Finally, we remark that the unlearning for generative models is still in its emerging stage; we do hope our works could contribute to a better setup/definition for this problem space.
>
>
>
> [1] Chourasia, Rishav, and Neil Shah. "Forget Unlearning: Towards True Data-Deletion in Machine Learning." ICML 2023. https://icml.cc/virtual/2023/poster/23753
>
>
> [2] Li, Tianhong, et al. "Mage: Masked generative encoder to unify representation learning and image synthesis." Proceedings of the IEEE/CVF Conference on Computer Vision and Pattern Recognition. 2023.
>
> [3] He, Kaiming, et al. "Masked autoencoders are scalable vision learners." Proceedings of the IEEE/CVF conference on computer vision and pattern recognition. 2022.
>
> [4] Saharia, Chitwan, et al. "Palette: Image-to-image diffusion models." ACM SIGGRAPH 2022 Conference Proceedings. 2022.

---

> > ### Author Response · Authors · 2023-11-16
> > **Response to Reviewer L8mN (2/2)**
> >
> > **Question 2**: The proposed approach itself is not too novel, while the work may have applied it first on image-to-image generative models. This approach can be thought of as a simple extension student-teacher or continual learning framework, and similar ideas have been explored in classification literature [5,6].
> >
> >
> > **Response 2**:
> > Indeed, our approach is inspired by the widely used teacher-student paradigm for machine unlearning. However, previous student-teacher based approach has some drawbacks for unlearning image-to-image generative models [5,6]:
> >
> > - Previous unlearning approaches directly compute the KL-divergence between the final outputs of the teacher and  student networks [5,6]; this is feasible and reasonable for classification tasks since the outputs are exactly single-variable discrete distributions $y$ with $c$ possible values, where $c$ is the number of classes for a given classfication task. Specifically, given input image $X$, teacher network $f_T: \mathcal{X}\to \Delta_c$ and student network $f_S: \mathcal{X}\to \Delta_c$, we can directly compute $D_{KL}(f_T(X)||f_S(X))$ since $f_T(X)$ and $f_S(X)$ are already distributions after the SoftMax function.
> >
> > However, since the image-to-image model generates high-dimensional images, directly computing the KL-divergence for the high-dimensional distribution is extremely expensive and not practical [7]. More precisely, given input image $X$, teacher network $f_T$ and student network $f_S$, the output $f_T(X)$ (or $f_S(X)$)  is an image; we need to estimate the distribution for each pixel first and the joint distribution of all possible pixels combinations.
> > As a contrast, our rigorous and novel theoretical analysis proves that the intractable KL-divergence computation can be efficiently approximated by a MSE-loss on the low-dimensional representation space. This is one of the major contributions to our approach (cf. *Machine unlearning.* in Page 2 and *Efficient Unlearning Approach.* in Page 6).
> >
> >
> >
> > - Previous unlearning approaches don’t have a thorough analysis for the solution space thus typically proposing some ad-hoc methods without concrete theoretical support [5,6]. Instead, we prove that Gaussian noise is the unique optimal solution for the unlearning on image-to-image generative models. This is a fundamental difference from previous works.
> >
> > - Previous unlearning approaches are highly coupled with the classification models; this is relatively easy to handle since typically people use cross-entropy to train the model. However, different image-to-image generative models use distinct loss functions. Therefore, we aim to find a universal solution for different types of models instead of proposing ad-hoc solutions for each type of models. Fortunately, our theoretical analysis is applicable to all image-to-image generative models with encoder-decoder architecture. That is, the MSE-loss on the representation space is generally applicable regardless of their model types or training loss functions.
> >
> >
> > We hope our explanation has made the novelty of our approach clearly emphasized.
> >
> > **Question 3**: The experimental setup for retain sample availability is not quite clear to me. Why are retain samples selected from the remaining 800 random classes? Also, this confused me if all the 1000 classes, or only 200 classes are used for the main experiments to train the original model.
> >
> >
> > **Response 3**: First, out of 1000 classes from ImageNet-1K, our experimental setup selects 100 classes as the retain set $D_R$ and another 100 classes as the forget set $D_F$; the remaining 800 classes $D_O$ are not considered either as retain or forget. We evaluate our methods with two different setups:
> >
> >
> > - Use the images from exact 100 classes from $D_R$ to do the unlearning. This is what we called *real retain samples*.
> >
> > - Suppose we don’t have access to the images from 100 classes from $D_R$, we use images from the other 800 classes $D_O$ to serve as the proxy retain set.
> >
> > For both cases, the images from the exact classes of $D_F$ is needed. Moreover, the performance/test on the retain set is always conducted on the 100 classes from $D_R$.
> >
> >
> > As for the original model, we download the open-source models pre-trained on ImageNet-1K as our starting point. We then fine-tune these models on these selected classes; this is the *original model* we used in this paper.
> >
> >
> >
> > Thanks again and we would be happy to provide more clarifications and answer any follow-up questions.
> >
> >
> >
> >
> > [5] Zhang, Xulong, et al. "Machine Unlearning Methodology base on Stochastic Teacher Network." arXiv preprint arXiv:2308.14322 (2023).
> >
> > [6] Zhang, Yongjing, et al. "Machine Unlearning by Reversing the Continual Learning." Applied Sciences 13.16 (2023): 9341.
> >
> > [7] Belghazi, Mohamed Ishmael, et al. "Mutual information neural estimation." International conference on machine learning. PMLR, 2018.

---

> ### Author Response · Authors · 2023-11-22
> **Kindly expecting further discussions**
>
> Dear Reviewer L8mN,
>
> Thanks again for constructive feedback on our paper. We hope that our previous responses have successfully addressed your concerns. We would like to remind you that the author-reviewer discussion period is closing very soon. If there are any remaining questions or concerns, we are looking forward to further discussions!
>
> Best regards,
>
> The Authors of Paper 2859

---

### Official Review · Reviewer_uD61 · 2023-11-05

**Soundness:** 2 fair
**Presentation:** 3 good
**Contribution:** 2 fair
**Rating:** 5
**Confidence:** 4

**Summary:**

This paper presents a study on machine unlearning for image-to-image generative models, an area not extensively explored previously. The paper introduces a theoretically sound and computationally efficient algorithm for unlearning that ensures minimal impact on the performance of retained data while effectively removing information from data meant to be forgotten. Demonstrated on ImageNet-1K and Places-365 datasets, the algorithm uniquely operates without needing the retained data, aligning with stringent data privacy policies. The research claims to pioneer a theoretical and practical approach to machine unlearning in the context of generative models.

**Strengths:**

Strengths:
- The paper boasts a clear and logical structure, helping readers' comprehension of the concepts presented.
- It ventures into the relatively untapped domain of applying machine unlearning to image-to-image (I2I) generative tasks.
- The authors have bolstered their approach with a solid theoretical foundation, enhancing the credibility and robustness of their proposed method.

**Weaknesses:**

Weakness:
- The study does not align with the foundational concept of machine unlearning, which typically necessitates a comparison between the unlearned and retrained models as per references [1-7]. Although the authors justify this divergence due to the high costs associated with retraining generative models, this deviates from the core goal of machine unlearning aimed at addressing privacy concerns. Particularly, the approach presented in this paper generates conspicuous Gaussian noise over the 'forgotten' data, which may inadvertently signal that the data was previously part of the training set, contradicting privacy preservation goals. A more compelling motivation might be found in text-to-image (T2I) scenarios [8] where the goal is to prevent the generation of inappropriate content, or in image-to-image (I2I) applications [9] that showcase practical utility. It would be more meaningful—and privacy-compliant—if the model could reconstruct unremarkable images that don't trace back to the original training data, rather than reconstruct images with evident distortions signaling prior data use.
- The evaluation process presented in the paper is incomplete with regard to established machine unlearning protocols. Typically, an unlearned model's performance is assessed using three distinct datasets: a test dataset to determine its generalization capability, a retained dataset to evaluate performance on non-forgotten data, and a forget dataset to check the efficacy of the unlearning process. The paper's Table 1 appears to only present results for the latter two, omitting the crucial evaluation on the general test dataset.
- The consideration of relevant baselines in the paper is lacking. Reference [9] describes unlearning in the context of image-to-image (I2I) generative models, which appears to be closely related to the work at hand. A comparison or a clear explanation of why the methods from [9] cannot integrate into the proposed framework would strengthen the current approach by situating it within the broader research landscape and justifying its unique contributions.
- The scope of the study with respect to the application of machine unlearning in image-to-image (I2I) generative models appears to be inaccurately broad. The term "machine unlearning for I2I generative models" suggests a wide range of applications; however, the paper primarily focuses on the image inpainting task. It would be more precise to either expand the variety of I2I applications examined in the study or to specifically define the scope as "machine unlearning for image inpainting tasks" to reflect the content more accurately. This would ensure clarity in the paper's contributions and avoid overgeneralization of the results.


>[1] Graves, Laura, Vineel Nagisetty, and Vijay Ganesh. "Amnesiac machine learning." Proceedings of the AAAI Conference on Artificial Intelligence. Vol. 35. No. 13. 2021.
>
>[2] Chundawat, Vikram S., et al. "Zero-shot machine unlearning." IEEE Transactions on Information Forensics and Security (2023).
>
>[3] Chen, Min, et al. "Boundary Unlearning: Rapid Forgetting of Deep Networks via Shifting the Decision Boundary." Proceedings of the IEEE/CVF Conference on Computer Vision and Pattern Recognition. 2023.
>
>[4] Warnecke, Alexander, et al. "Machine unlearning of features and labels." arXiv preprint arXiv:2108.11577 (2021).
>
>[5] Jia, Jinghan, et al. "Model sparsification can simplify machine unlearning." arXiv preprint arXiv:2304.04934 (2023).
>
>[6] Kurmanji, Meghdad, Peter Triantafillou, and Eleni Triantafillou. "Towards Unbounded Machine Unlearning." arXiv preprint arXiv:2302.09880 (2023).
>
>[7] Golatkar, Aditya, Alessandro Achille, and Stefano Soatto. "Eternal sunshine of the spotless net: Selective forgetting in deep networks." Proceedings of the IEEE/CVF Conference on Computer Vision and Pattern Recognition. 2020.
>
>[8] Gandikota, Rohit, et al. "Erasing concepts from diffusion models." arXiv preprint arXiv:2303.07345 (2023).
>
>[9] Moon, Saemi, Seunghyuk Cho, and Dongwoo Kim. "Feature unlearning for generative models via implicit feedback." arXiv preprint arXiv:2303.05699 (2023).

**Questions:**

- Could you please add one baseline mentioned in the weakness to the paper (Table 1)?
- Why easing concepts can be thought about as a noisy label method? Please give more explanations.

---

> ### Author Response · Authors · 2023-11-18
> **Response to Reviewer uD61(1/3)**
>
> We thanks for Reviewer's feedback and comments. Please see our responses to your questions below.
>
>
> **Question 1** The study does not align with the foundational concept of machine unlearning, which typically necessitates a comparison between the unlearned and retrained models as per references [1-7].
>
>
> **Response 1** This is a fundamental question, and we understand the reviewer's concern about the definition/setup of our unlearning approach. Indeed, we did ask the same questions ourselves. First, for *classifications tasks*, we agree that most of previous work considers that the unlearned model should be indistinguishable to the model retrained only on the retain set (golden baseline). However, there are three points we want to bring to this reviewer’s attention:
>
>
> - First, even the indistinguishability to the golden standard may have some issues. Indeed, recent work has shown that *‘some unlearning algorithms can generate models indistinguishable from the golden baseline beyond any arbitrarily small threshold while still exposing the deleted data’*[10]. In other words, the indistinguishability of training on the retain set alone *cannot* guarantee the information is indeed removed from the model.
>
> - Second, retraining these generative models from scratch on the retain set is beyond our (and average user, in general) computational capabilities. More precisely, for the three models evaluated in our paper, it would take more than 10K GPU hours ($>$50 days on our 8-GPU server; estimated from the experimental setup in [11,12,13]) to train the model from scratch; this is clearly infeasible given the resources we have at hand and the tight deadline for this rebuttal period.
>
>
>
> - Given these considerations, our setup/definition primarily focuses on the practical scenarios of copyright protection and/or pornographic/violence content control, other than the indistinguishability from the retrained model. For example, if the original training set for the original model got some pornographic images by accident, our method can make the unlearned model not generate those pornographic images in a very efficient way (as opposed to expensively retraining the model from scratch). Moreover, the results show that on the retain set, our approach has negligible performance drop compared to the model before unlearning. Consequently, we believe our approach is sound for copyright protection and pornographic/violence content control hence very useful in real applications.
>
> **Question 2**: Although the authors justify this divergence due to the high costs associated with retraining generative models, this deviates from the core goal of machine unlearning aimed at addressing privacy concerns. Particularly, the approach presented in this paper generates conspicuous Gaussian noise over the 'forgotten' data, which may inadvertently signal that the data was previously part of the training set, contradicting privacy preservation goals. A more compelling motivation might be found in text-to-image (T2I) scenarios [8] where the goal is to prevent the generation of inappropriate content, or in image-to-image (I2I) applications [9] that showcase practical utility. It would be more meaningful—and privacy-compliant—if the model could reconstruct unremarkable images that don't trace back to the original training data, rather than reconstruct images with evident distortions signaling prior data use.
>
> **Response 2**:  This is a very insightful comment (*the approach presented in this paper generates conspicuous Gaussian noise over the 'forgotten' data, which may inadvertently signal that the data was previously part of the training set, contradicting privacy preservation goals.*). We note that, under our setup, given an image with noise distortions, one cannot conclude whether it is in the training set or not. For example, suppose our method makes the model forget the concept *car*; given any image of a car, the corresponding generated image typically will have noise distortions, but it is possible that this specific image is not used for training. Even though one can infer that some concepts are from the forget set, that still meets our main goal, i.e., the unlearned model does not generate the unwanted images.

---

> > ### Author Response · Authors · 2023-11-18
> > **Response to Reviewer uD61(2/3)**
> >
> > **Question 3**: Could you please add one baseline mentioned in the weakness to the paper? The consideration of relevant baselines in the paper is lacking. A comparison or a clear explanation of why the methods from [9] cannot integrate into the proposed framework would strengthen the current approach.
> >
> > **Response 3**: We appreciate for pointing out these papers. As we mention in the paper, our approach pioneers the unlearning for Image-to-Image generative models, so there are no previous baselines we can directly compare against. Indeed, we tried our best to make comparisons with the unlearning approaches for classification tasks that are either directly applicable or modifiable for I2I generative tasks.
> >
> > - Some of the baseline methods mentioned by the reviewer are actually already included in our paper and compared with our approach. Specifically, the approaches presented in [4,8] are simply about maximizing the training loss on the forget set.  This is precisely what we call *Max Loss* baseline in our paper. For reviewer's convenience, we reproduce the portion of our results for diffusion models on image inpainting tasks:
> >
> > |8$\times$8 patches|FID: $D_R\downarrow$|FID: $D_F\uparrow$|IS: $D_R\uparrow$|IS: $D_F\downarrow$|CLIP: $D_R\uparrow$|CLIP: $D_F\downarrow$|
> > |:-:|:-:|:-:|:-:|:-:|:-:|:-:|
> > |Original model|12.2|14.6|19.3|23.1|0.88|0.89|
> > |Max Loss|34.1|45.7|12.8|17.1|0.77|0.76|
> > |Ours|**13.4**|**107.9**|**19.4**|**10.3**|**0.87**|**0.69**|
> >
> >
> > |4$\times$4 patches|FID: $D_R\downarrow$|FID: $D_F\uparrow$|IS: $D_R\uparrow$|IS: $D_F\downarrow$|CLIP: $D_R\uparrow$|CLIP: $D_F\downarrow$|
> > |:-:|:-:|:-:|:-:|:-:|:-:|:-:|
> > |Original model|7.8|6.0|10.3|11.2|0.93|0.96|
> > |Max Loss|11.9|15.4|10.0|11.0|0.88|0.93|
> > |**Ours**|**8.2**|**39.8**|**10.3**|**10.7**|**0.93**|**0.88**|
> > > **NOTE:** '$\uparrow$' means higher is better and '$\downarrow$' means lower is better.
> >
> >
> > As shown above, our approach consistently beats Max Loss under different metrics. Specifically, our approach preserves well the knowledge on the retain set; our approach only introduces a negligible performance drop compared to the original model. In contrast, the Max Loss approach has a remarkable performance drop on the retain set. As for the forget set, our approach does remove more information compared to the MAX Loss method. For more results, please check Table 1 and Table 2 (page 7) and Figures D.8, D.9, D.10, and D.11 (page 24-27) in our paper.
> >
> >
> > - As for the baseline methods non-included in our reference list, please see our rationale below:
> >     - [1] requires the storage of the gradient for each parameter of every training step when training the original models. This approach is not scalable given the extremely large training set and the enormous model size for the latest image generative models.  For example, given the VQ-GAN we use in our paper, the original model has 176M parameters which are trained with 400K steps; hence, we need to store $176M\times 400K=7.04\times 10^{14}=70.4T$ gradients values [11]; such high-capacity hardware is not available for practical application.
> >     - [2,5] are orthogonal to our approach. [2] focuses on the data access issues of *existing* unlearning algorithms and suggests using the images generated by the original model as an alternative training set. Besides, the main contribution of [5] is that they observe that pruning the original model before unlearning can improve the performance of many *existing* unlearning algorithms. In contrast, our approach focuses on designing a *new* unlearning algorithm, which is orthogonal to these two works.
> >     - [3] is specially designed for classification tasks based on the concept of *Decision Boundary* between different classes; thus, it is not applicable or modifiable to generative models.
> >     - [6] is also highly coupled with classification tasks and requires the exact KL-divergence between output distributions of different models; thus, it is not applicable or modifiable to image-to-image generative models since computing KL-divergence on high-dimensional images are mathematically not practical.
> >     - [7] requires computing the NTK for the entire training set used to train the original model; the total number of elements in NTK is given by $(m\times |D|)^2$, where $m$ is the output dimensions and $|D|$ is the number of training samples. Given the ImageNet dataset with 256x256 resolution used to train the VQ-GAN model, the total number of elements in corresponding NTK is $(256\times 256\times3 \times 1.28M)^2\approx 6.33\times 10^{18}$ [11]; such huge amounts of memory costs make it not doable in practice.
> >     - [9] is not applicable to diffusion models. Besides, as shown in Figure 3 of [9], [9] targets the unlearning of some simple *binary* targeted feature, e.g., the hairstyle with or without bangs. In contrast, our approach is general so it can work with MAE, GAN and diffusion models; moreover, our approach targets a more complex setup with hundreds of images classes.

---

> ### Author Response · Authors · 2023-11-18
> **Response to Reviewer uD61(3/3)**
>
> **Question 4**: The evaluation process presented in the paper is incomplete with regard to established machine unlearning protocols. Typically, an unlearned model's performance is assessed using three distinct datasets: a test dataset to determine its generalization capability, a retained dataset to evaluate performance on non-forgotten data, and a forget dataset to check the efficacy of the unlearning process. The paper's Table 1 appears to only present results for the latter two, omitting the crucial evaluation on the general test dataset.
>
> **Response 4**: This is a very important question. We would like to clarify that we *actually performed* such a test (the first evaluation in the reviewer's question) in our paper, but simply described it in a slightly different way.
>
> Specifically, out of 1000 classes from ImageNet-1K, our experimental setup selects 100 classes as the retain set $D_R$ and another 100 classes as the forget set $D_F$; the remaining 800 classes $D_O$ are not considered either as retain or forget sets. We evaluate our method with the following setup: In Table 1 and  Table 2 (rows *Proxy* $D_R$) of our paper, we assume that we don’t have access to the images from 100 classes from $D_R$, and then we use images from the other 800 classes $D_O$ to serve as the proxy retain set for our unlearning approach. Moreover, the performance/test on the retain set is always conducted on the 100 classes from original $D_R$.
>
> In fact, for this setup, we can re-consider the original $D_R$ as the *general test set* and re-consider the $D_O$ as the new $D_R$. As shown in Table 1 and Table 2 (rows *Proxy* $D_R$), our approach is evaluated on the 100 classes from *general test set* and it shows negligible performance drop on the general test set.
>
>
> **Question 5**: The scope of the study with respect to the application of machine unlearning in image-to-image (I2I) generative models appears to be inaccurately broad. The term "machine unlearning for I2I generative models" suggests a wide range of applications; however, the paper primarily focuses on the image inpainting task. It would be more precise to either expand the variety of I2I applications examined in the study or to specifically define the scope as "machine unlearning for image inpainting tasks" to reflect the content more accurately. This would ensure clarity in the paper's contributions and avoid overgeneralization of the results.
>
>
> **Response 5**: We appreciate this reviewer's suggestion. We will modify our title to 'Machine Unlearning for Image Completion Generative Tasks'.
>
>
> **Question 6**: Why easing concepts can be thought about as a noisy label method? Please give more explanations.
>
> **Response 6**: the reason we use Gaussian noise as the optimization target for the forget set is motivated by our theoretical analysis. Specifically, we prove that Gaussian noise is the unique optimal solution for unlearning on image-to-image generative models if we want to destroy the performance on the forget set. This is indeed one of the main contributions of this paper.
>
> Thanks again for your comments; we would be happy to provide more clarifications and answer any follow-up questions.
>
> [1] Graves, Laura, et al. "Amnesiac machine learning." AAAI 2021.
>
> [2] Chundawat, Vikram S., et al. "Zero-shot machine unlearning." IEEE Transactions on Information Forensics and Security (2023).
>
> [3] Chen, Min, et al. "Boundary Unlearning: Rapid Forgetting of Deep Networks via Shifting the Decision Boundary." CVPR  2023.
>
> [4] Warnecke, Alexander, et al. "Machine unlearning of features and labels." arXiv preprint arXiv:2108.11577 (2021).
>
> [5] Jia, Jinghan, et al. "Model sparsification can simplify machine unlearning." arXiv preprint arXiv:2304.04934 (2023).
>
> [6] Kurmanji, Meghdad, et al. "Towards Unbounded Machine Unlearning." arXiv preprint arXiv:2302.09880 (2023).
>
> [7] Golatkar, Aditya, et al. "Eternal sunshine of the spotless net: Selective forgetting in deep networks." CVPR  2020.
>
> [8] Gandikota, Rohit, et al. "Erasing concepts from diffusion models." arXiv preprint arXiv:2303.07345 (2023).
>
> [9] Moon, Saemi, et al. "Feature unlearning for generative models via implicit feedback." arXiv preprint arXiv:2303.05699 (2023).
>
> [10] Chourasia, Rishav, et al. "Forget Unlearning: Towards True Data-Deletion in Machine Learning." ICML 2023. https://icml.cc/virtual/2023/poster/23753
>
> [11] Li, Tianhong, et al. "Mage: Masked generative encoder to unify representation learning and image synthesis." CVPR  2023.
>
> [12] He, Kaiming, et al. "Masked autoencoders are scalable vision learners." CVPR 2022.
>
> [13] Saharia, Chitwan, et al. "Palette: Image-to-image diffusion models." ACM SIGGRAPH 2022 Conference Proceedings. 2022.

---

> > ### Comment · Reviewer_uD61 · 2023-11-20
> > **Thank you for the clarifications!**
> >
> > Thank you for your detailed responses and clarifications to my previous queries. I appreciate the effort you've put into addressing each point, which has largely resolved my initial questions and concerns. As a result, I am inclined to raise my score. However, I would like to offer some additional suggestions and points for consideration.
> >
> > Regarding Q1:
> > I understand and acknowledge your motivations concerning the challenges of retraining, which are central to your paper's premise. However, I still hold reservations about the first point in your response as a robust support for your method. While I appreciate the practical implications of your approach, I believe that a more solid theoretical foundation or guarantee that your method effectively removes the targeted information from the model would strengthen your argument. Furthermore, showcasing a practical application, as hinted at in your motivation, could significantly enhance the paper's impact and relevance. Demonstrating this application would not only align with your motivations but also provide tangible evidence of your method's effectiveness.
> >
> > Regarding Q3:
> > Your response here is well-received. Discussing the distinctions between machine unlearning (MU) in classification and generation contexts is crucial, and your explanations are compelling. The comparison of methods between these contexts enriches the paper’s content. However, I noticed that reference [5], which proposes adding a sparse regularization term to the general machine unlearning objective function, seems like it could be directly adapted to a more general case. (Maybe need to be double-checked by authors). I suggest adding this part to the revision. Including this analysis would provide a more comprehensive view of how your method situates within the broader landscape of MU techniques.

---

> ### Author Response · Authors · 2023-11-22
> **Follow-up discussion (1/2)**
>
> Dear Reviewer uD61,
>
> We thank you very much for raising the rating for our paper. We appreciate your new comments and suggestions very much. Please see our responses to your new questions below:
>
> **Question**: Regarding *Q1*: I understand and acknowledge your motivations concerning the challenges of retraining, which are central to your paper's premise. However, I still hold reservations about the first point in your response as a robust support for your method. While I appreciate the practical implications of your approach, I believe that a more solid theoretical foundation or guarantee that your method effectively removes the targeted information from the model would strengthen your argument. Furthermore, showcasing a practical application, as hinted at in your motivation, could significantly enhance the paper's impact and relevance. Demonstrating this application would not only align with your motivations but also provide tangible evidence of your method's effectiveness.
>
>
> **Response**: Ideally, for a given unlearned model, the generated images ($\hat{X}_F$) on the forget set should have no information related to the corresponding ground truth images ($X_F$). From the perspective of mutual information (MI), this means that the mutual information between $\hat{X}_F$ and $X_F$ is 0, i.e., $I(X_F;\hat{X}_F)=0$.
>
> In our paper, our theoretical analysis and proposed optimization methods aim to remove the related information from the generated images ($\hat{X}_F$) by gradually maximizing its mutual information w.r.t a Gaussian Noise. To further verify the effectiveness, we estimate the mutual information between the reconstructed images and the corresponding ground truth images on both forget set and retain set.
>
> We note that estimating the mutual information for high-dimensional images is extremely difficult. To this end, we follow the standard practice and use one of the state-of-the-art approximate estimation methods which compute a variational lower bound of exact mutual information [14]. Moreover, we also compute the variational mutual information bound between the ground truth images and Gaussian noise for a reference. The results are shown below:
>
> |MI Estimator| SMILE_1.0 | SMILE_5.0 |
> |:-:|:-:|:-:|
> | Original Model: $I(X_F;\hat{X_F})$  | 10.95     | 12.20     |
> | Unlearned Model: $I(X_F;\hat{X_F})$ | 5.68      | 5.79      |
> | $I(X_F;\mathcal{N})$                | 4.24      | 6.04      |
> | Original Model: $I(X_R;\hat{X_R})$  | 12.54     | 16.98     |
> | Unlearned Model: $I(X_R;\hat{X_R})$ | 12.83     | 17.05     |
> | $I(X_R;\mathcal{N})$                | 5.29      | 4.42      |
>
> > **NOTE:**  SMILE_1.0 and SMILE_5.0 are using the same SMILE mutual information estimator but with different hyper-parameters [14].
>
>
> As shown above, after unlearning on the forget set, the mutual information between the reconstructed images ($\hat{X}_F$) and the corresponding ground truth images ($X_F$) $I(X_F;\hat{X}_F)$ is much lower than the original model (i.e., before unlearning). Moreover, after unlearning, $I(X_F;\hat{X}_F)$ is very close to the $I(X_F;\mathcal{N})$.
>
> As for the retain set, the $I(X_R;\hat{X}_R)$ is roughly the same as before unlearning, which means that the information on the retain set is well preserved.
>
> Finally, per reviewer’ suggestion, we plan to add a companion demo for the final version of our paper; this demo will show our approach in action on a few generic applications, hence illustrate our method’s effectiveness.

---

> ### Author Response · Authors · 2023-11-22
> **Follow-up discussion (2/2)**
>
> **Question**: Regarding *Q3*: Your response here is well-received. Discussing the distinctions between machine unlearning (MU) in classification and generation contexts is crucial, and your explanations are compelling. The comparison of methods between these contexts enriches the paper’s content. However, I noticed that reference [5], which proposes adding a sparse regularization term to the general machine unlearning objective function, seems like it could be directly adapted to a more general case. (Maybe need to be double-checked by authors). I suggest adding this part to the revision. Including this analysis would provide a more comprehensive view of how your method situates within the broader landscape of MU techniques.
>
> **Response**: Indeed, we mentioned that [5] is orthogonal to our approach. To further demonstrate this point, we combine our approach together with the $L1$-regularization techniques introduced in [5]. We use the so called '*linear decaying*' trick on page 6 of [5] and compare the performance with and without $L1$-regularization for VQ-GAN, under the same setup mentioned in Section 4.1. Due to time limitations, we conduct the test only on a subset of the validation set with 5 images per class instead of the entire set (50 images/class). The results are shown below:
>
>
> | 4$\times$4 Patches | FID: $D_R\downarrow$ | FID: $D_F\uparrow$ | IS: $D_R\uparrow$ | IS: $D_F\downarrow$ | CLIP: $D_R\uparrow$ | CLIP: $D_F\downarrow$ |
> |:-:|:-:|:-:|:-:|:-:|:-:|:-:|
> | Ours (No $L1$) | 73.7 | 73.2 | 12.1 | 11.4 | 0.84 | 0.76 |
> | Ours+$L1$ | 73.7 | 73.0 | 12.2 | 11.1 | 0.84 | 0.76 |
>
> | 8$\times$8 Patches | FID: $D_R\downarrow$ | FID: $D_F\uparrow$ | IS: $D_R\uparrow$ | IS: $D_F\downarrow$ | CLIP: $D_R\uparrow$ | CLIP: $D_F\downarrow$ |
> |:-:|:-:|:-:|:-:|:-:|:-:|:-:|
> | Ours (No $L1$) | 85.3 | 142.2 | 13.0 | 9.6 | 0.75 | 0.57 |
> | Ours+$L1$ | 85.8 | 139.7 | 12.9 | 9.6 | 0.74 | 0.57 |
>
> As shown above, by combining our approach with $L1$-regularization from [5], the results are roughly the same as the case without $L1$-regularization.
>
> We also tried different coefficients $\lambda$ for the $L1$-regularization. The above results are the coefficients that achieve optimal trade-off between the performance on the retain set and forget set. We found that the performance on the retain set is pretty sensitive to the value of $\lambda$; specifically, the performance on the retain set will have a significant drop if the $\lambda$ is relatively large (e.g., $>10^{-6}$). We have the following comment regarding this observation:
>
> - Previous work reveals that *sparsity of networks is relatively high for fairly simple and naive tasks (e.g., CIFAR-10/100, ImageNet, GLUE, etc.)*[15]. In other words, if the network is designed for more challenging tasks such as image generation, then there is less sparsity inside the network compared to the classification tasks. This may be the reason why $L1$-regularization from [5] doesn't achieve the same improvements in image generative tasks as in classification tasks.
>
> Thank you again for your constructive feedback. We would be happy to provide more clarifications and answer any follow-up questions.
>
> [5] Jia, Jinghan, et al. "Model sparsification can simplify machine unlearning." arXiv preprint arXiv:2304.04934 (2023).
>
> [14] Song, Jiaming, et al. "Understanding the limitations of variational mutual information estimators." ICLR 2020.
>
> [15] Liu, Shiwei, et al. "Sparsity May Cry: Let Us Fail (Current) Sparse Neural Networks Together!." ICLR 2023.

---

### Comment · Reviewer_hK5e · 2023-11-19
**Review stop**

Hello everybody,

on Friday I had an emergency medical intervention on my eyes and I am now unable to read for the next few weeks. This message is also written with an assistance.

Overall, I cannot further contribute to the review process.

All the best

---

> ### Author Response · Authors · 2023-11-22
> **Thanks again for the reviewing efforts!**
>
> Dear Reviewer hK5e,
>
> We are sorry to hear this from your side. We thank you again for your efforts and engagement for the review process of our paper. Wishing you a speedy and smooth recovery.
>
> Take care and all the best.
>
> Best,
>
> The Authors of paper 2859

---

### Meta-Review · Area_Chair_wU6D · 2023-12-06

**Metareview:**

This paper tackles of the problem of machine unlearning for generative models. To accomplish this, they propose an optiomization problem which aims to achieve low KL-divegence for the retain set, and maximize the KL-diveregnce for the forget set. Overall, the writing and flow the paper is appropriate and the approach is intuitive. While reviewers have concerns regarding the practicality/usage cases of the evaluated setting. Look at the rebuttal, I believe those concerns have been addressed.

**Justification For Why Not Higher Score:**

While the paper is sufficient for publication, the proposed approach and the result are some what expected.

**Justification For Why Not Lower Score:**

I believe the reviewers' concerns were addressed. The paper tackles an interesting problem with a quite reasonable solution.

---

### Decision · Program_Chairs · 2024-01-16

Accept (poster)